

# Opening Pandora's box: How to constrain regional projections of the carbon cycle

Lina Teckentrup[1,2], Martin G. De Kauwe[3], Gab Abramowitz[1,2], Andrew J. Pitman[1,2], Anna M. Ukkola[1,2], Sanaa Hobeichi[1,2], Bastien François[4], and Benjamin Smith[5,6]

[1]ARC Centre of Excellence for Climate Extremes, Sydney, NSW, Australia
[2]Climate Change Research Centre, University of New South Wales, Sydney, NSW, Australia
[3]School of Biological Sciences, University of Bristol, England
[4]Laboratoire des Sciences du Climat et l'Environnement (LSCE-IPSL) CNRS/CEA/UVSQ, UMR8212, Université Paris-Saclay, Gif-sur-Yvette, France
[5]Hawkesbury Institute for the Environment, Western Sydney University, Penrith, NSW, Australia
[6]Department of Physical Geography and Ecosystem Science, Lund University, Lund, Sweden

**Correspondence:** Lina Teckentrup (l.teckentrup@unsw.edu.au)

**Abstract.** Climate projections from global circulation models (GCMs) part of the Coupled Model Intercomparison Project 6 (CMIP6) are often employed to study the impact of future climate on ecosystems. However, especially at regional scales, climate projections display large biases in key forcing variables such as temperature and precipitation, which hamper predictive capacity. In this study we examine different methods to constrain regional projections of the carbon cycle in Australia. We employ a dynamic global vegetation model (LPJ-GUESS) and force it with raw output from CMIP6 to assess the uncertainty associated with the choice of climate forcing. We then test different methods to either bias correct or calculate ensemble averages over the original forcing data to constrain the uncertainty in the regional projection of the Australian carbon cycle. We find that all bias correction methods reduce the bias of continental averages of steady-state carbon variables. Carbon pools are insensitive to the type of bias correction method applied for both individual GCMs and the arithmetic ensemble average across all corrected models. None of the bias correction methods consistently improve the change in carbon over time, highlighting the need to account for temporal properties in correction or ensemble averaging methods. Some bias correction methods reduce the ensemble uncertainty more than others. The vegetation distribution can depend on the bias correction method used. We further find that both the weighted ensemble averaging and random forest approach reduce the bias in total ecosystem carbon to almost zero, clearly outperforming the arithmetic ensemble averaging method. The random forest approach also produces the results closest to the target dataset for the change in the total carbon pool, seasonal carbon fluxes, emphasizing that machine learning approaches are promising tools for future studies.

## 1 Introduction

Global circulation models (GCMs) are useful projection tools of future climate at continental and global scales and above but inevitably simulate large biases in temperature, precipitation and humidity at regional scales and at individual grid points (Randall et al., 2007; Flato et al., 2013). Projections of atmospheric variables from GCMs, represented by the Coupled Model



Intercomparison Project (CMIP), underpin a suite of critical future predictions of the carbon and water cycles (e.g. Ahlström et al., 2012; Ukkola et al., 2016; Ahlström et al., 2017), species distributions (Cheaib et al., 2012), species resilience to climate extremes (Sperry et al., 2019) and predictions of conservation planning (Gallagher et al., 2021). Critically, many applications utilise atmospheric variables from GCMs as forcing without explicitly considering underlying uncertainty in their

(bias-corrected) climate projections. This uncertainty includes, but is by no means limited to, the fact that CMIP is an 'ensemble of opportunity', and not explicitly designed to represent an independent set of estimates, i.e. CMIP models share modules and are related to varying degrees (e.g. Annan and Hargreaves, 2017; Boe, 2018; Abramowitz et al., 2019).

To tackle biases in GCM forcing a range of approaches have been employed, with no clear agreement or 'best practice' on how to assess GCM skill and to bias correct simulated climate variables, and/or to weight ensemble members. Some studies

have quantified the sensitivity of impact studies to GCM selection method, the choice of bias correction, and/or the ensemble averaging techniques. For example, Gohar et al. (2017) examined the impact of bias correction methods on future warming levels and found that both selecting GCMs based on performance and bias correcting model data reduced uncertainties in regional projections. In an Australian study, Johnson and Sharma (2015) increased model consensus in future drought projections using bias corrected simulations. These studies focused either directly on the climate variables and/or derived relatively

simple indices based on a single variable. In an analysis of hazard indices based on multiple climate drivers, Zscheischler et al. (2019) showed multivariate methods tended to outperform univariate bias-correction methods. In addition, Kolusu et al. (2021) tested the impact of different weighting techniques and two bias correction methods on the spread of hydrological risk profiles and found that the sensitivity to climate model weighting was considerably smaller than the uncertainty resulting from bias correction methodologies. When Ahlström et al. (2012) used CMIP5 simulations to run the dynamic global vegetation model

(DGVM) LPJ-GUESS, they found that GCM climate biases translated into a divergence in the future simulated (offline) carbon cycle responses on regional and global scales that was significantly reduced when the climatological input forcing was bias corrected (Ahlström et al., 2017). The need to address biases in GCM forcing is commonly acknowledged, but the wide range in possible solutions (e.g., bias correction, ensemble averages across GCMs) makes it difficult to determine the impact of the correction in climate forcing on the specific question of interest.

There have been multiple efforts to constrain future multi-model ensemble uncertainty (e.g. Michelangeli et al., 2009; Knutti et al., 2010b; Bárdossy and Pegram, 2012; Bishop and Abramowitz, 2013; Johnson and Sharma, 2015; François et al., 2020). Most of these attempts assume that the GCMs that simulate the historical climate well are likely to provide more skillful future projections. Based on this assumption, different approaches for dealing with ensemble uncertainty have emerged that can broadly be grouped into three strategies: (i) selecting only a subset of GCMs fit for the respective study (e.g. Pennell and

Reichler, 2011; Rowell et al., 2016; Herger et al., 2018; Gershunov et al., 2019); (ii) applying downscaling and bias correction methods (e.g. Panofsky et al., 1958; Wood et al., 2004; Déqué, 2007; Michelangeli et al., 2009; Bárdossy and Pegram, 2012; François et al., 2020); and/ or (iii) applying ensemble weighting techniques (e.g. Bishop and Abramowitz, 2013; Sanderson et al., 2017; Massoud et al., 2019, 2020).

The first strategy focuses on sub-selecting GCMs from the full ensemble, using metrics deemed to be application relevant, to

obtain an ensemble that is truly representative of the uncertainty linked to GCM simulations. Commonly, this is based on how





well GCMs simulate relevant climate variables compared to historical observations (e.g. Kolusu et al., 2021) and represents the 'skilled models' category, shown in figure 1. Other studies find that excluding the 'weakest' models has little impact on the overall uncertainty range (e.g. Déqué and Somot, 2010; Knutti et al., 2010b; Rowell et al., 2016). Some studies choose models defined as independent (e.g. based on the correlation of the biases in the simulations or within a Bayesian framework;

Jun et al., 2008; Knutti et al., 2010a; Pennell and Reichler, 2011; Annan and Hargreaves, 2017). Lastly, Evans et al. (2014) and Cannon (2015) suggest selecting those models that 'span' the (plausible) CMIP projections when selecting GCMs for dynamical downscaling ('bounding' models category in fig. 1).

The second strategy employs a range of bias correction methods to reduce errors in the GCM outputs. Univariate bias correction methods are widely used to improve agreement of the statistical attributes (mean, variance, quantiles) of the simulated

climate variables with those of historical climate data. While these methods can produce reasonable results (e.g. Yang et al., 2015; Casanueva et al., 2018) they typically correct each climate variable independently, one grid cell at a time. This can result in inconsistent relationships across physically interlinked climate variables, and/or across a spatial domain. Given univariate methods do not account for multidimensional dependencies, they cannot correct temporal, inter-variable or spatial aspects of the simulations (François et al., 2020). To address these gaps, multivariate methods account for dependencies between variables

and spatial patterns. Multivariate methods are especially valuable in impact modeling frameworks where the combination of atmospheric processes across a range of time and space scales, such as coinciding low rainfall and high temperatures inducing vegetation drought stress, are important (Zscheischler et al., 2019).

Finally, several weighting methods have been developed to derive ensemble averages. The arithmetic multi-model mean is commonly used (Knutti et al., 2010a) and by cancelling non-systematic errors, usually out-performs individual GCMs.

However, assigning each ensemble member a uniform weight has been criticised (Knutti et al., 2010b; Herger et al., 2019). Non-uniform weights, based on skill, independence, or skill and independence combined (e.g. Bishop and Abramowitz, 2013; Brunner et al., 2019, 2020) can also be used. In addition, machine learning techniques have become increasingly popular to calculate multi-model averages (e.g. Huntingford et al., 2019; Thao et al., 2022) that use GCM outputs as predictors to match an observation based target (e.g. reanalysis products). For example, Wang et al. (2018) explored a random forest approach,

support vector machine, and Bayesian model averaging to calculate a best-fit multi-model ensemble average for monthly temperature and precipitation over Australia. Similarly, other studies have focused on climate extremes (e.g. Deo and Şahin, 2015; Yunjie Liu et al., 2016) and climate impacts on the environment (e.g. Jung et al., 2010; Yang et al., 2016; Wu et al., 2019a) using machine learning approaches.

In this study, we focus on Australia, and analyse the impact of climate forcing bias correction and ensemble averaging meth-

ods on the simulated historical carbon cycle. Australia is a suitable study system for this work because climate projections of precipitation will remain uncertain at regional scales for the foreseeable future (IPCC, 2013; Ukkola et al., 2020; Grose et al., 2020) and are likely to have a disproportionate influence on water-limited regions such as Australia, with potential impacts on vegetation distributions, and water and carbon cycles, given many biologically relevant processes are threshold-based and disproportionately responsive to extremes as opposed to mid-range changes in climate forcing. Recently, Teckentrup et al. (2021)

showed that 13 dynamic global vegetation models from the TRENDY project (Friedlingstein et al., 2019) simulated markedly



different magnitudes of net biome productivity, resulting in a significant divergence in the long-term historical accumulated vegetation carbon stock (-4.7 to 9.5 Pg C $\text{yr}^{-1}$). These differences in carbon accumulation occurred despite the use of a single, common meteorological forcing which underlines the urgent need to better constrain uncertainty in climate forcing to facilitate robust assessments of the future terrestrial carbon cycle demonstrated for Australia as a case study. Here, we assess the impact

of different CMIP6 GCM selection, bias correction and ensemble averaging methods on the simulated carbon cycle. We use a single dynamic global vegetation model, LPJ-GUESS (Smith et al., 2014), and focus on responses at seasonal to centennial timescales. LPJ-GUESS is the only second-generation DGVM part of the TRENDY ensemble, and can therefore be expected to simulate more realistic temporal carbon dynamics than first-generation DGVMs (e.g. Fisher et al., 2018). Our goal is to examine how the choice of method to deal with CMIP6 model uncertainty influences the projection of the terrestrial carbon

cycle and whether any selected method represents a robust or preferable choice.

## 2 Future climate forcing

### 2.1 CMIP6

We chose 21 CMIP6 GCMs (see tab. 1) that provide the three meteorological forcing variables needed to run LPJ-GUESS, i.e. the near-surface air temperature (tas), the total precipitation flux (pr) and the incoming shortwave radiation (rsds), and

examine the r1i1p1f1 realisation that covers the time period (1850–2100). Four GCMs (ACCESS-CM2, ACCESS-ESM1-5, BCC-CSM2-MR and NESM3) provide incoming shortwave radiation starting in 1950 only. For these GCMs, we recycled incoming shortwave radiation of the first 25 years of the available forcing (i.e. 1950–1974) for the first 100 years (i.e. 1850–1949). All GCMs provide daily data but differ in their spatial resolution. We therefore regridded all GCMs to a common 0.5° grid using first order conservative remapping to match the resolution of the reanalysis and the native grid of LPJ-GUESS, and

focus on the historical time period (1901-2019).



**Table 1.** CMIP6 models used to force LPJ-GUESS. Further details for each model are available at the references listed in this table.

| GCM | Institute ID | Native resolution (lat × lon) | Key reference |
|---|---|---|---|
| ACCESS-CM2 | CSIRO-ARCCSS | 1.25° × 1.875° | Bi et al. (2013) |
| ACCESS-ESM1-5 | CSIRO | 1.25° × 1.875° | Law et al. (2017) |
| BCC-CSM2-MR | BCC | 1.121° × 1.125° | Wu et al. (2019b) |
| CanESM | CCCma | 2.7905° × 2.8125° | Swart et al. (2019) |
| CESM2-WACCM | NCAR | 1.3° × 0.9° | Liu et al. (2019) |
| CMCC-CM2-SR | CMCC | 0.94° × 1.25° | Cherchi et al. (2019) |
| EC-Earth | EC-Earth-Consortium | ∼0.7° × 0.7° | Döscher et al. (2022) |
| EC-Earth3-Veg | EC-Earth-Consortium | ∼0.7° × 0.7° | Döscher et al. (2022) |
| GFDL-CM4 | NOAA-GFDL | 1° × 1.25° | Held et al. (2019) |
| GFDL-ESM4 | NOAA-GFDL | 1° × 1.25° | Dunne et al. (2020) |
| INM-CM4-8 | INM | 1.5° × 2° | Volodin et al. (2018) |
| INM-CM5-0 | INM | 1.5° × 2° | Volodin et al. (2018) |
| IPSL-CM6A-LR | IPSL | 1.3° × 2.5° | Boucher et al. (2020) |
| KIOST-ESM | KIOST | 1.875° × 1.875° | Pak et al. (2021) |
| MIROC6 | MIROC | 1.4° × 1.4° | Tatebe et al. (2019) |
| MPI-ESM1-2-HR | MPI-M | 0.94° × 0.94° | Mauritsen et al. (2019), Müller et al. (2018) |
| MPI-ESM1-2-LR | MPI-M | 1.865° × 1.875° | Mauritsen et al. (2019) |
| MRI-ESM2-0 | MRI | 1.121° × 1.125° | Yukimoto et al. (2019) |
| NESM3 | NUIST | 1.865° × 1.875° | Cao et al. (2018) |
| NorESM2-LM | NCC | 1.9° × 2.5° | Seland et al. (2020) |
| NorESM2-MM | NCC | 0.94° × 1.25° | Seland et al. (2020) |

## 2.2 Historical climate forcing

We chose the CRUJRA reanalysis product (Harris, 2019) as the reference dataset to compare with the unconstrained CMIP6 results, as well as to derive bias corrections and ensemble weights. CRUJRA is derived from the Climatic Research Unit gridded Time Series (CRU TS) v4.03 monthly data (Harris et al., 2014) and from the Japanese 55-year Reanalysis data (JRA-55) (Kobayashi et al., 2015). Temperature, downward solar radiation flux, specific humidity and precipitation in JRA-55 are aligned to temperature, cloud fraction, vapour pressure and precipitation in CRU TS (v4.03), respectively. The CRUJRA dataset spans the years 1901–2018 on a 6 hour timestep which we aggregated to a daily temporal resolution, at a 0.5° spatial resolution.



## 2.3 Dataset sensitivity

The CRUJRA reanalysis is not "observations" and, as with all reanalyses, is subject to uncertainty itself. To test the sensitivity
to the choice of reference dataset, we compared the CRUJRA to the ERA5 reanalysis dataset.

ERA5 is the fifth generation reanalysis from the European Centre for Medium-Range Weather Forecasts (ECMWF; Hers-
bach et al., 2020). It uses a linearized quadratic 4D-var assimilation scheme that takes the timing of the observations and model
evolution within the assimilation window into account. Compared to the predecessor ERA-Interim reanalysis, it has a higher
spatiotemporal resolution and assimilates more observations. The reanalysis is produced at an hourly time step and covers the
time period 1979–2020. Its horizontal resolution is $0.05°$. As for the CRUJRA reanalysis, we aggregated the data to a daily
timestep and regridded the dataset to a $0.5°$ spatial resolution using first-order conservative regridding.

## 3  Methods

To assess the sensitivity of carbon cycle projections to different GCM selection, bias correction and ensemble averaging meth-
ods, we followed the steps outlined in figure 1 and detailed below.

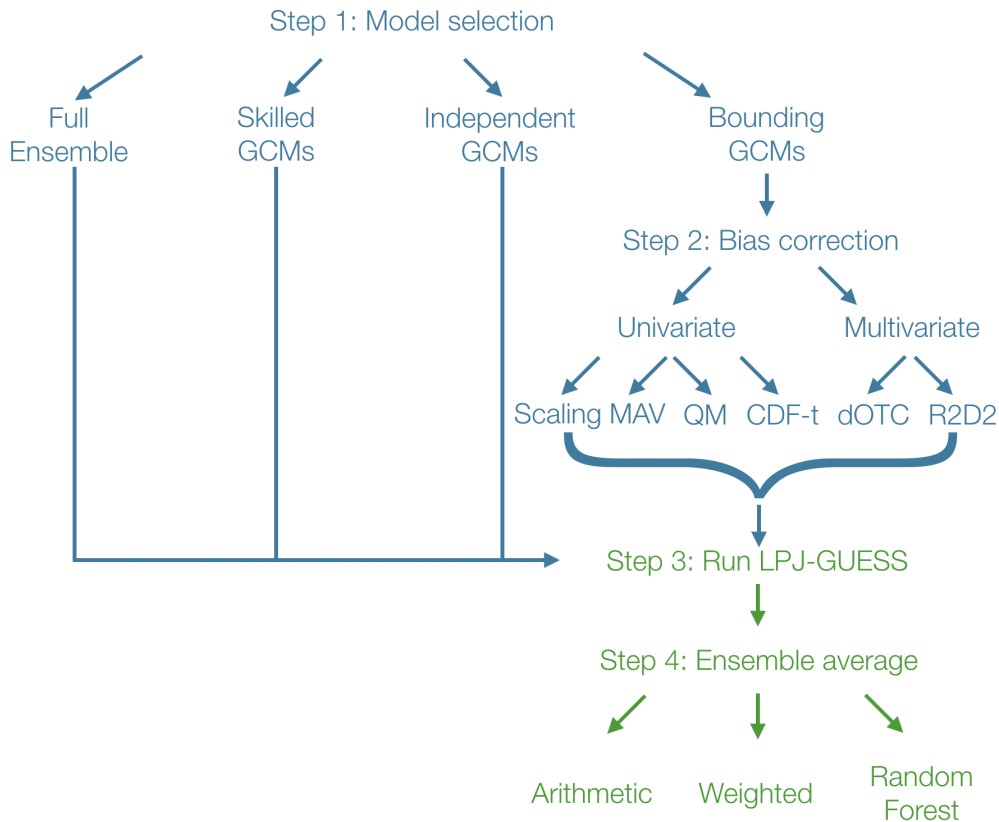

**Figure 1.** Schematic for study set-up. All terms are defined in the text and the key steps are described in the text.



## 3.1 Step 1: Model selection

Our first step was to decide whether to use the full CMIP6 ensemble ('Full ensemble') or to select a subset of GCMs based on a selection criterion ('skilled', 'independent', 'bounding', see fig. 1 step 1 and appendix fig. A1). Since precipitation is the single largest driver of variability in the Australian carbon cycle (Haverd et al., 2013), we selected the GCMs solely based on the performance of projected precipitation. We next describe each of the selection criteria in more detail (see fig. 1 step 1).

### 3.1.1 Skill

An intuitive way to select CMIP GCMs is to define a set of performance metrics and select those GCMs with a pre-defined level of skill (e.g. Rowell et al., 2016; Gershunov et al., 2019). We calculated the metrics suggested by Haughton et al. (2018) (see tab. 2) using the CRUJRA reanalysis as the reference dataset for daily, monthly and annual precipitation, then ranked all GCMs for each metric and finally chose the GCMs with the highest average rank for monthly and annual timescales. For the last method (overlap of histogram), we estimated the intervals ('bin size') using the Freedman Diaconis Estimator for the reference dataset (CRUJRA) and then used the same bin size for the simulated variable (i.e. CMIP forcing).

**Table 2.** Metrics used to evaluate GCM performance (compare Haughton et al., 2018). O is the observation, here the reanalysis, and S is the simulation.

| Metric | Formulation |
|--------|-------------|
| Root mean squared error | $\sqrt{\frac{\sum(O_i - S_i)^2}{n}}$ |
| Normalised Mean Error | $\frac{\sum |O_i - S_i|}{\sum |O_i - \bar{O}|}$ |
| Mean bias error | $\sum \frac{S_i - O_i}{n}$ |
| Difference in standard deviation | $|1 - \frac{\sigma_S}{\sigma_O}|$ |
| Correlation | $corr(O, S)$ |
| Difference in 5th percentile | $P_5(S) - P_5(O)$ |
| Difference in 95th percentile | $P_{95}(S) - P_{95}(O)$ |
| Difference in skewness | $|1 - \frac{skew(S)}{skew(O)}|$ |
| Difference in kurtosis | $|1 - \frac{kurt(S)}{kurt(O)}|$ |
| Overlap of histogram | $\sum(min(bin_{S,k}, bin_{O,k}))$ |

### 3.1.2 Independence

The CMIP6 ensemble is not designed to be an ensemble of independent models, and therefore there is a risk that the members of the ensemble share systematic biases. We therefore seek to select GCMs that are independent of each other, in order to obtain a better sample of model projections. Here we defined that GCMs are independent if their (here: precipitation) biases are uncorrelated with any of the other ensemble members. We derived the bias by subtracting the reanalysis from the simulated precipitation and then calculated the Pearson correlation coefficient between the different CMIP6 GCMs on monthly and annual





timescales and and chose the GCMs with a weak correlation coefficient (i.e. lower than 0.3; compare Bishop and Abramowitz, 2013).

### 3.1.3 Bounding models

Similar to Evans et al. (2014), we also chose GCMs that span the largest range of simulated precipitation based on the average, the interannual variability (IAV) and the change of average precipitation in the last 30 years of the historical time period (1989–2018) compared to 1901–1930. Accordingly, the five bounding GCMs are the driest (INM-CM4-8) and the wettest (MPI-ESM1-2-HR) GCM, the GCMs with the lowest (KIOST-ESM) and highest (NorESM-MM) IAV in precipitation and the GCMs with the lowest (EC-Earth3-Veg) and the highest (NorESM2-MM) change of average precipitation in 1989–2018 relative to the 1901–1930 average.

### 3.2 Step 2: Bias correction methods

Once a selection of GCMs is made, the biases of a given GCM can be corrected (see fig. 1 step 2). We explored six approaches using CRUJRA as our reference dataset. We corrected the three climate forcing variables, i.e. temperature, precipitation and incoming shortwave radiation, and derived the correction based on the calibration time period 1989–2010 given this is common to both reanalysis products used here. We applied each method per pixel so that the different grid points were corrected independently of each other and tested the correction on both daily and monthly timescales. We show the corrections based on daily timescales in the main figures, and use the corrections based on monthly timescales to assess the sensitivity to the correction timescale in the supplement. To understand the sensitivity to the correction technique, we only corrected the five bounding models (see section 3.1.3) because they defined the total CMIP6 ensemble spread. In the subsections below, we describe the methods in more detail. Let define $O$ and $S$ the observed and simulated variables at the same grid point for the calibration time period. $P$ is the simulated variable for the projection period to adjust with bias correction methods, and $C$ is the resulting bias-corrected variable. The projection period was split into ten 25-year slices. The bias correction was then derived and applied to each calendar month within each time slice separately. Let $P_t$ and $C_t$ being the values of the variables at time $t$.

*Univariate*
Univariate bias correction methods are applied independently to each forcing variable and grid cell.

### 3.2.1 Scaling

We calculated additive (temperature) and multiplicative (precipitation and incoming shortwave radiation) scaling bias corrections based on the 1989–2010 climatology (compare e.g. Chen et al., 2011). For temperature, the bias-corrected value at time $t$ for the projection period is derived as follows:

$$C_t = P_t - \overline{S} + \overline{O}, \tag{1}$$



with $\overline{S}$ and $\overline{O}$ the means of the variables $S$ and $O$, respectively. For precipitation and incoming shortwave radiation, bias-corrected values are derived according to

$$C_t = \frac{P_t}{\overline{S}} \cdot \overline{O}. \tag{2}$$

to avoid negative values.

### 3.2.2   Mean and variance correction (MAV)

Here, we aimed to additionally correct the variance in the temperature forcing. We followed equation 1 and accounted for the variance by multiplying by the ratio between the standard deviation of the observed and simulated variables $\sigma_O$ and $\sigma_S$. The forcing variables are corrected following

$$C_t = (P_t - \overline{S}) \cdot \frac{\sigma_O}{\sigma_S} + \overline{O}. \tag{3}$$

We used the precipitation and incoming shortwave radiation corrected following the multiplicative correction (see eqn. 2) since the (proportional) scaling correction affects both mean and variance.

### 3.2.3   Quantile mapping (QM)

We employed the univariate quantile mapping (QM) method (Panofsky et al., 1958; Wood et al., 2004; Déqué, 2007) which adjusts the cumulative distribution function of a modeled climate variable to that of the observed one. Let denote $F_O$ and $F_S$ the cumulative distribution function (CDF) of the observed and simulated variables. By linking CDFs between the model and the reference, the QM method allows to derive the bias-corrected value $C_t$ as follows:

$$C_t = F_O^{-1}(F_S(P_t)), \tag{4}$$

where $F_O^{-1}$ is the inverse cumulative distribution function of $O$.

### 3.2.4   Cumulative Distribution Function (CDF-t)

The 'Cumulative Distribution Function – Transform' (CDF-t; Michelangeli et al., 2009) is a version of quantile mapping that adjusts the cumulative distribution function of the simulated climate variables using a quantile-mapping transfer function. The difference with QM is that, by linking cumulative distribution functions using a two-step procedure, CDF-t is specifically designed to take into account the simulated changes of CDFs from the calibration to the projection period. Thus, that the future climate scenarios incorporate the model's projected changes in both mean climate and variability at all time scales up to the decadal. More details can be found in (Vrac et al., 2012). Implementing the CDF-t method in the present study in addition to the QM method would allow to assess the influence of taking into account simulated distribution changes in the bias correction procedure on results of regional projections of carbon cycle.



 *Multivariate*

As opposed to univariate bias correction methods, multivariate bias correction methods are able to take inter-variable dependencies into account. Here each multivariate bias correction method is applied independently at each grid cell to jointly adjust temperature, precipitation, and incoming shortwave radiation. By doing so, the multivariate bias correction methods are aimed to correct inter-variable dependencies within each grid cell.

### 3.2.5 Dynamical Optimal Transport Correction (dOTC)

The 'dynamical Optimal Transport Correction' method (dOTC, Robin et al., 2019) is a generalization of the CDF-t method to the multivariate case. By using optimal transport theory, dOTC is designed to adjust both univariate distributions and dependence structures of the simulated variables. Moreover, following the philosophy of CDF-t, dOTC is able not only to preserve the simulated changes in the univariate distributions between the calibration and the projection periods but also the simulated change in multivariate properties (e.g., induced by climate change). For more details and equations, see Robin et al. (2019); François et al. (2020).

### 3.2.6 Rank Resampling For Distributions and Dependences (R2D2)

The 'Rank Resampling For Distributions and Dependences' method ('R2D2', Vrac, 2018) is based on the Schaake Shuffle (Martyn Clark et al., 2004). The Schaake Shuffle is a reordering technique that reorders a sample so that its rank structure corresponds to the rank structure of a reference sample. This allows the reconstruction of multivariate dependence structures. As a first step, the R2D2 performs the univariate CDF-t bias correction (see 3.2.4). The method allows for the possibility to select a 'reference dimension' for the Schaake Shuffle, i.e., one physical variable at one given site, for which rank chronology remains unchanged. The reconstruction of inter-variable correlations of the reference is then performed using the Schaake Shuffle with the constraint of preserving the rank structure for the reference dimension. For more details and equations, see Robin et al. (2019); François et al. (2020).

## 3.3 Step 3: Run LPJ-GUESS

We ran LPJ-GUESS with a reference dataset (CRUJRA reanalysis), the full raw CMIP6 ensemble (which includes the skilled, independent and bounding models) and additionally with the bounding models (see section 3.1.3) after they were bias corrected according to the methods 3.2.1–3.2.6.

LPJ-GUESS (Smith et al., 2014, Lund–Potsdam–Jena General Ecosystem Simulator; ) is a widely used dynamic global vegetation model for climate–carbon studies (Sitch et al., 2003; Smith et al., 2014). LPJ-GUESS simulates the exchange of water, carbon and nitrogen through the soil–plant–atmosphere continuum (Smith et al., 2014) by accounting for resource competition for light and space between plants. We adopted the global configuration of the model that uses 12 plant functional types (PFTs), simulating differences in growth form (grasses, broadleaved trees or deciduous trees), photosynthetic pathway (C3 or C4), phenology (evergreen, summer green or rain green), tree allometry, life history strategy, fire sensitivity, and bioclimatic



limits for establishment and survival (see Smith et al., 2014, for details). LPJ-GUESS is the only second-generation DGVM part of the TRENDY ensemble (compare Fisher et al., 2010, 2018) and explicitly represents demographic processes, such as stand age/size structure development, mortality and competition among locally co-occurring PFT populations, as well as disturbance-induced heterogeneity across the landscape of a grid cell.

240     We use LPJ-GUESS version 4.0.1 in 'cohort mode', where woody plants of the same size and age co-occur in a 'patch' and as such, are represented by a single average individual. Each PFT is represented by multiple average individuals, and one PFT cohort is defined as the average of several individuals. We run LPJ-GUESS with the plant and soil nitrogen dynamics switched on. Fire is simulated annually (stochastically) based on temperature, fuel availability and the moisture content of upper soil layer as a proxy for litter moisture content (Thonicke et al., 2001).

## 3.4 Step 4: Ensemble averages

After running LPJ-GUESS with either the raw or corrected climate data (step 3), the final step was to calculate an ensemble average of the resulting carbon fluxes. We focussed on the total carbon storage ($C_{Total}$) and foliar projective cover (FPC) over Australia at annual timesteps, and the gross primary productivity (GPP) at seasonal timesteps. We explored three different approaches based on the full ensemble or the selected models (see section 3.1)

### 3.4.1 Arithmetic ensemble average

We first calculated the arithmetic ensemble average where each of the GCM+LPJ-GUESS ensemble members was assigned the same weight.

### 3.4.2 Skill and independence

Following Bishop and Abramowitz (2013), we calculated weights based on both independence and skill. We here chose the carbon variables resulting from the reference LPJ-GUESS run (driven with the CRUJRA reanalysis) as the target variable, and the carbon variables resulting from the LPJ-GUESS runs forced with the CMIP6 as the predictor variables. This method accounts for both the performance differences and their error dependencies. In a first step, the bias with respect to observational data is calculated. The method then uses the error correlation coefficient as a metric for error dependencies. This method derives the linear combination of the CMIP6 members to minimise the mean square difference to the results from the reanalysis runs following:

$$C_w^j = w^T x^j = \sum_{k=1}^{K} w_k x_k^j \qquad (5)$$





where $j$ represent the grid cells, and $k$ is the number of the ensemble members. Consequently, $x_k^j$ is the value of the k$^{\text{th}}$ bias-corrected model (i.e., after subtracting the mean error from the dataset) at the $j^{th}$ grid cell. The weights ($w^T$) provide an analytical solution to the minimization of

$$\sum_{j=1}^{J} (C_w^j - x_{obs}^j)^2 \qquad (6)$$

when subject to the constraint that the sum of the weights ($w_k$) always adds up to 1. The solution can be expressed as:

$$w = \frac{\mathbf{A^{-1}1}}{\mathbf{1^T A^{-1} 1}} \qquad (7)$$

where $\mathbf{1^T} = \overbrace{[1, 1, ..., 1]}^{k\,elements}$ and $\mathbf{A}$ is the K × K difference covariance matrix.

### 3.4.3 Random Forest

Random forest is an ensemble learning method that constructs a collection of decision trees and then outputs a weighted average of predictions of the individual trees. For each decision tree, a subset of training samples are randomly selected following a bootstrap sampling approach. At each node, a random sample of predictor variables is selected for splitting. We varied the number of predictor variables and number of trees, and here show the results that produced the lowest error. The metric of splitting is the sum of squares of errors. As in method 3.4.2, we chose the carbon variables resulting from the reference
LPJ-GUESS run (driven with the CRUJRA reanalysis) as the target variable, and the carbon variables resulting from the LPJ-GUESS runs forced with the CMIP6 as the predictor variables. We further included the latitude and longitude as predictors, and when analysing monthly data, the month. The random selections change as the 'tree' grows following a random sampling with the replacement approach. The algorithms involved in different decision trees are run in parallel. Both the random sampling procedure and the parallelism in algorithm operations mean that the predictor blocks in random forest are built independently.

### 3.5 Summary of methods

Our methods examine many of the approaches previously used to select from and/or constrain the CMIP6 ensemble in carbon cycle modelling. While not all possible combinations of approaches were examined, we employed a wide range of methods. In this study, we seek to examine how applying these corrections methods affect the simulation of the Australian carbon cycle by LPJ-GUESS as a case study. In the following, we use the abbreviations defined in table 3.





**Table 3.** List of LPJ-GUESS runs and ensemble averaging methods tested in this study.

| Run | LPJ-GUESS forced with | Ensemble abbreviation | Averaging method | Based on |
|---|---|---|---|---|
| $LG_{CRUJRA}$ | CRUJRA reanalysis | $ENS_{Arithmetic,Full}$ | Arithmetic | Full CMIP6 ensemble (raw) |
| $LG_{EC-Earth3-Veg}$ | Raw EC-Earth3-Veg climate | $ENS_{Arithmetic,Skill}$ | Arithmetic | Skilled GCMs (raw) |
| $LG_{INM-CM4-8}$ | Raw INM-CM4-8 climate | $ENS_{Arithmetic,Independence}$ | Arithmetic | Independent GCMs (raw) |
| $LG_{KIOST-ESM}$ | Raw KIOST-ESM climate | $ENS_{Arithmetic,Bounding}$ | Arithmetic | Bounding GCMs (raw) |
| $LG_{MPI-ESM1-2-HR}$ | Raw MPI-ESM1-2-HR climate | $ENS_{Arithmetic,Bounding,Scaling}$ | Arithmetic | Corrected bounding GCMs (scaling) |
| $LG_{NorESM2-MM}$ | Raw NorESM2-MM climate | $ENS_{Arithmetic,Bounding,MAV}$ | Arithmetic | Corrected bounding GCMs (MAV) |
| | | $ENS_{Arithmetic,Bounding,QM}$ | Arithmetic | Corrected bounding GCMs (QM) |
| | | $ENS_{Arithmetic,Bounding,CDF-t}$ | Arithmetic | Corrected bounding GCMs (CDF-t) |
| | | $ENS_{Arithmetic,Bounding,R2D2}$ | Arithmetic | Corrected bounding GCMs (R2D2) |
| | | $ENS_{Arithmetic,Bounding,dOTC}$ | Arithmetic | Corrected bounding GCMs (dOTC) |
| | | $ENS_{Weighted}$ | Weighted | Full CMIP6 ensemble (raw) |
| | | $ENS_{RF}$ | Random forest | Full CMIP6 ensemble (raw) |





## 4 Results


We first examined the average and IAV (depicted by the standard deviation of the detrended annual precipitation and temperature) of the simulated and reanalysis annual precipitation and temperature over Australia between 1989–2018 (see fig. 2). Annual precipitation (1989–2018) simulated by the CMIP6 ensemble members varies widely from 254 mm yr$^{-1}$ (MPI-ESM1-2-HR) to 858 mm yr$^{-1}$ (INM-CM4-8). The CRUJRA reanalysis lies in the lower quartile of the CMIP6 spread (499

mm yr$^{-1}$, see fig. 2,c), implying a systematic over-estimate across the CMIP6 GCMs. The precipitation IAV varies between 55 mm yr$^{-1}$ (KIOST-ESM) and 183 mm yr$^{-1}$ (NorESM2-MM) and most CMIP6 ensemble members simulated higher IAV than the CRUJRA reanalysis (66 mm yr$^{-1}$; see fig. 2,c). Relative to 1901–1930, most CMIP6 GCMs do not show a significant trend (17 out of 21), two GCMs significantly increase in precipitation (up to 76 mm yr$^{-1}$ in the end of the historical time period; NorESM2-MM) and two GCMs significantly decrease (down to -59 mm yr$^{-1}$, EC-Earth3-Veg). CRUJRA slightly increases

in precipitation relative to 1901–1930 for the latter half of the historical time period (27.2 mm with a significant trend of 0.40 mm yr$^{-1}$; see fig. 2,d).





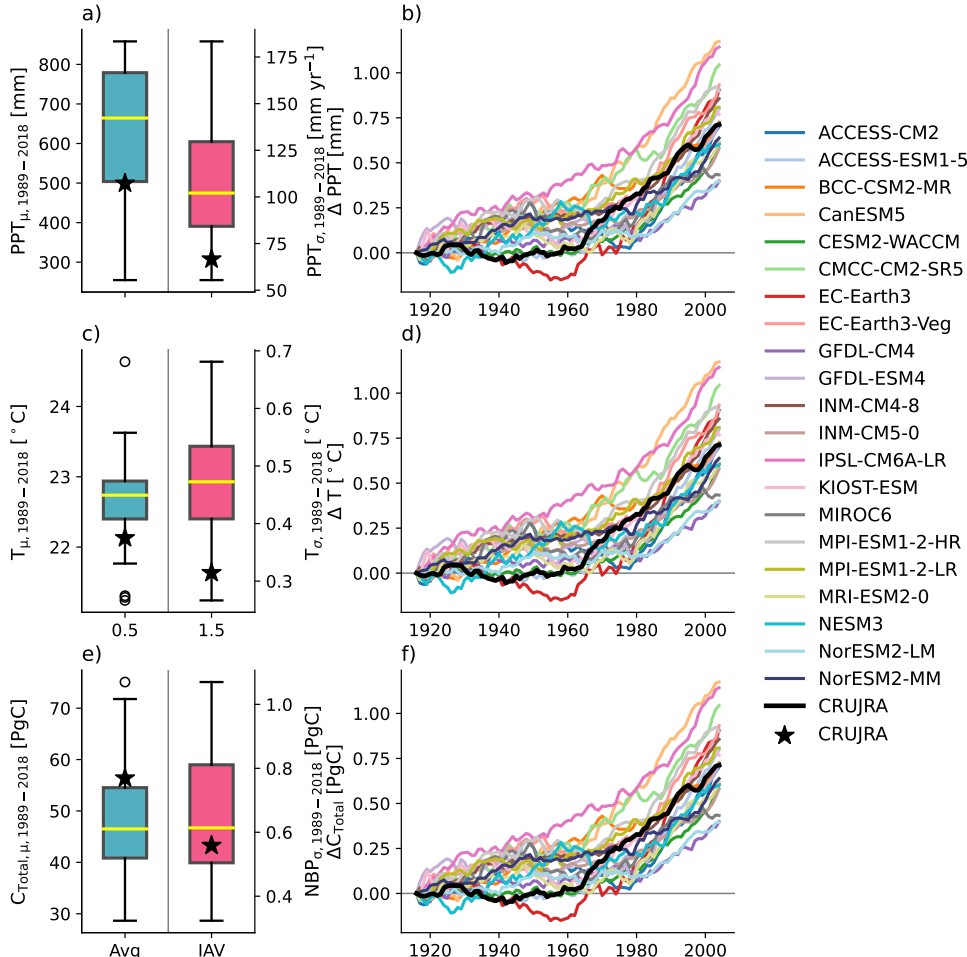

**Figure 2.** Average and interannual variability (IAV) of annual precipitation averaged over Australia for the time period 1989–2018 (a), average and IAV of annual temperature averaged over Australia for the time period 1989–2018 (c) for the 21 CMIP6 ensemble members (see tab. 1). Panel e shows the average of the total carbon stored in Australia for the time period 1989–2018 based on LPJ-GUESS simulations with the CMIP6 ensemble on the left and the IAV of the net biome productivity over Australia for the same time period on the right. The black stars represent the respective values obtained using the CRUJRA reanalysis. Panel b, d, and f show the 30-year moving average of the change of annual temperature, precipitation and total carbon storage respectively relative to the 1901–1930 average. The thick black line represents simulations obtained using the CRUJRA reanalysis.

The average simulated temperature over Australia for the last 30 years of the historical time period varies amongst the CMIP6 ensemble members from 21.2°C (INM-CM5-0) up to 24.6°C (MIROC6). The median of the full ensemble is 22.7°C and slightly higher than the average temperature for the CRUJRA reanalysis (22.1°C). The IAV in temperature ranges from 0.27°C (NorESM-LM) to 0.68°C (GFDL-ESM4). The CMIP6 GCMs tend to simulate higher IAV in temperature compared to






the year-to-year variability found in the CRUJRA reanalysis (0.31°C; see fig. 2, a). Relative to 1901–1930, all CMIP6 ensemble members show a continental average increases in temperature but to varying degrees (∼0.4–1.2°C averaged over 1989–2018; see fig. 2,b). We note that figure 2b, d, and f show the smoothed change in the according variable and do not allow conclusions on IAV.

Finally, figure 2 e, f show the impact of differences in the meteorological forcing on the average simulated total carbon pool ($C_{Total}$), the IAV in net biome productivity (NBP) and the change in $C_{Total}$ for Australia when LPJ-GUESS is forced with the raw climate forcing of each of the CMIP6 ensemble members. Depending on the choice of GCM, $C_{Total}$ varies between 28.6 PgC ($LG_{MPI-ESM1-2-HR}$) and 75.1 PgC ($LG_{INM-CM4-8}$). Compared to $C_{Total}$ simulated by $LG_{CRUJRA}$ (56.4 PgC), the LPJ-GUESS driven with CMIP6 forcing tends to simulate lower $C_{Total}$. The IAV in NBP ranges between 0.3 PgC

($LG_{KIOST-ESM}$) and 1.1 PgC ($LG_{CMCC-CM2-SR5}$). The IAV in NBP simulated by $LG_{CRUJRA}$ (0.6 PgC) falls into the lower interquartile range (IQR) of the CMIP6 ensemble runs. $C_{Total}$ for Australia increases by the end of the historical period for all CMIP6 forcings with values between 0.1 PgC ($LG_{EC-Earth3}$) and 4.1 PgC ($LG_{NorESM2-MM}$). Compared to the reanalysis results, most of the CMIP6 models lead to a weaker increase in $C_{Total}$ over the historical period (except for $LG_{INM-CM4-8}$, $LG_{INM-CM5-0}$, $LG_{NorESM2-LM}$, and $LG_{NorESM2-MM}$).

Taken together, figure 2 demonstrates both the uncertainties in meteorological variables obtained from GCMs and how these propagate to large simulation biases in Australia's carbon cycle. In the following, we examine the impact of correcting climate forcing on these biases.

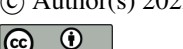



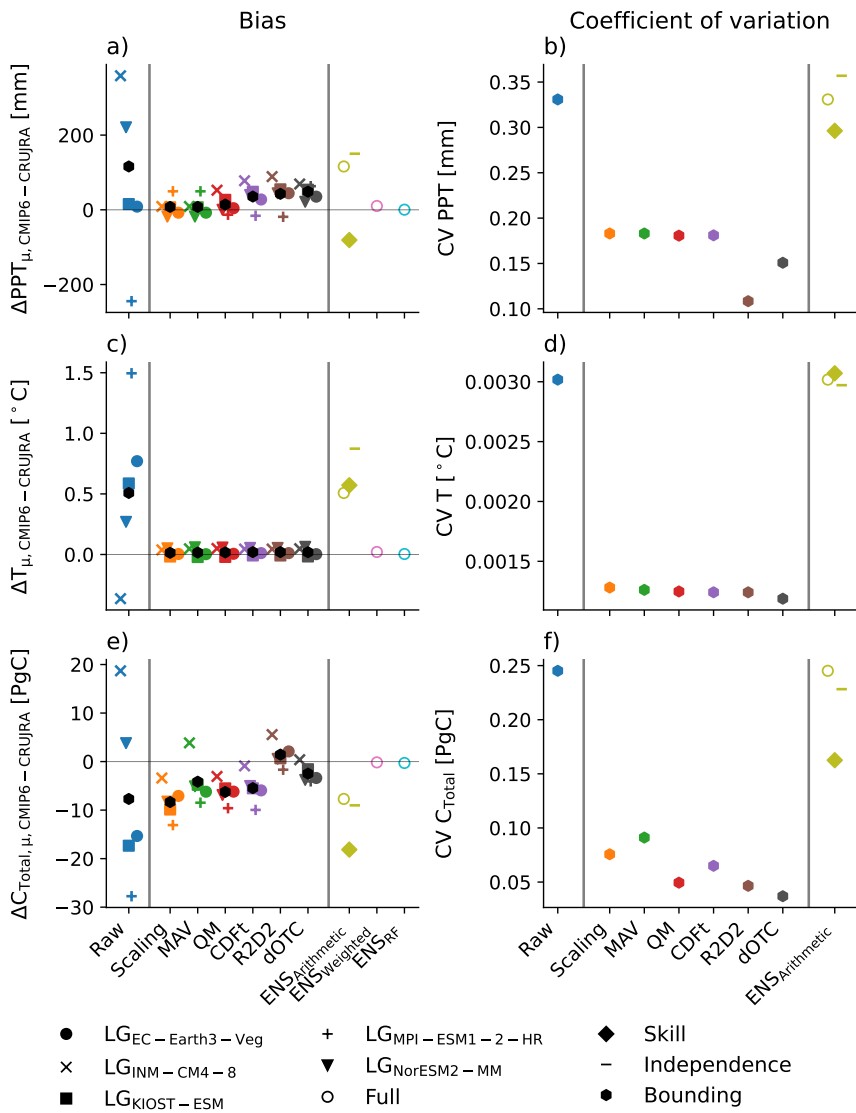

**Figure 3.** Difference between precipitation (PPT), temperature (T), and carbon storage ($C_{Total}$) based on the CMIP6 and CRUJRA forcing (a,c,e), and coefficient of variance across the ensemble of the same variables. The different colors represent the results based on the raw (blue) or corrected climate forcing using scaling (orange), mean and variance (MAV, green), quantile mapping (QM, red), cumulative distribution function - transform (CDF-t, purple), dynamical optimal transport correction (dOTC, brown), and matrix recorrelation (R2D2, dark grey) approaches and the three ensemble averaging methods (arithmetic mean (olive), weighted average (pink), and random forest (cyan)). The different symbols show LPJ-GUESS runs forced with the five bounding models EC-Earth3-Veg (filled circle), INM-CM4-8 (x), KIOST-ESM (square), MPI-ESM1-2-HR (+), and NorESM2-MM (triangle), the full ensemble (empty circle), and the three model selection methods skill (diamond), independence (horizontal bar), and bounding models (hexagon). The black hexagons depict the ensemble average of the LPJ-GUESS runs based on the raw and corrected bounding climate forcing.



The large ensemble spread in the CMIP6 forcing variables (see fig. 2 a–d) results in a large spread in the simulated carbon cycle (see fig. 2 e and f). Figure 3 a shows the biases in the forcing variables precipitation (PPT) and temperature (T) as well as $C_{Total}$ based on the CMIP6 compared to the results of the reanalysis. Positive values indicate that the results based on the CMIP6 forcing are higher compared to the reanalysis, and negative values demonstrate the opposite. Each of the bias correction methods reduces the bias in the forcing variables so that the bias in the corrected precipitation is significantly lower , and the bias in corrected temperature in comparison to the raw CMIP6 meteorology is close to zero (see fig. 3a,c). Consequently, $C_{Total}$ based on LPJ-GUESS driven with the corrected CMIP6 GCMs results in a smaller distance to $C_{Total}$ based on the $LG_{CRUJRA}$ run compared to the raw forcing for most LPJ-GUESS runs (see fig. 3 a). However, while the results based on the $LG_{NorESM2-MM}$ model initially simulated ∼3 PgC more than the runs based on the CRUJRA reanalysis, all univariate bias correction methods lead to larger biases from -5.0 PgC (CDF-t) to -8.3 PgC (Scaling) while the multivariate methods result in biases similar in magnitude (dOTC) or reduce it significantly (R2D2). When averages are calculated based on the full CMIP6 ensemble (hollow circles in fig. 3e), the random forest and weighted ensemble average approach produces almost identical results compared to the $LG_{CRUJRA}$ run (-0.29 PgC and -0.16 PgC, respectively; see fig. 3). The arithmetic ensemble average of $C_{Total}$ is with -7.7 PgC lower than the weighted average and the random forest approach. Figure 3e also shows the impact of model selection on calculated ensemble averages. Given both the weighted ensemble averaging and random forest approach are insensitive to redundant (i.e. models with similar biases) information we expect that testing those methods based on different GCM subsamples will yield similar results. We therefore only show the impact on the arithmetic average of $C_{Total}$. The values for the arithmetic average can depend on the selection of models it is derived from. Calculating the arithmetic average based on the full ensemble or on the five independent or bounding models gives similar results (but lower than the weighted and random forest approach: -9.0, and -7.6 PgC, respectively). Notably, the arithmetic ensemble average based on the five most skilled models produces the lowest value of all selection methods (-18.1 PgC). The arithmetic average of the bounding models is almost identical to that of the full ensemble for $C_{Total}$, and does not changes slightly with the correction method (black hexagons in fig. 3).

While the type of bias correction method only shows small alterations of the values of the arithmetic average of any of the variables examined in figure 3, the coefficient of variation (CV), which we here use as a measure for ensemble uncertainty, can vary depending on the method chosen. All bias correction methods reduce the CV compared to the raw CMIP6 data. For temperature, all bias correction methods result in similar values for CV (see fig. 3 d). Precipitation shows some variation depending on the type of bias correction method applied (univariate vs multivariate; see fig. 3 b). For temperature, the CV is robust and does not change strongly depending on the subselection of GCMs while for precipitation, selecting GCMs with high skill decreases the CV most. The CV of $C_{Total}$ is most reduced when the multivariate dOTC approach is applied on the forcing variables, and selecting the most skilled GCMs for an arithmetic average here yields the strongest reduction in CV compared to the full ensemble or selecting independent or bounding models.

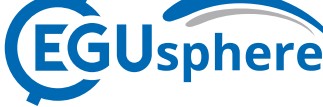

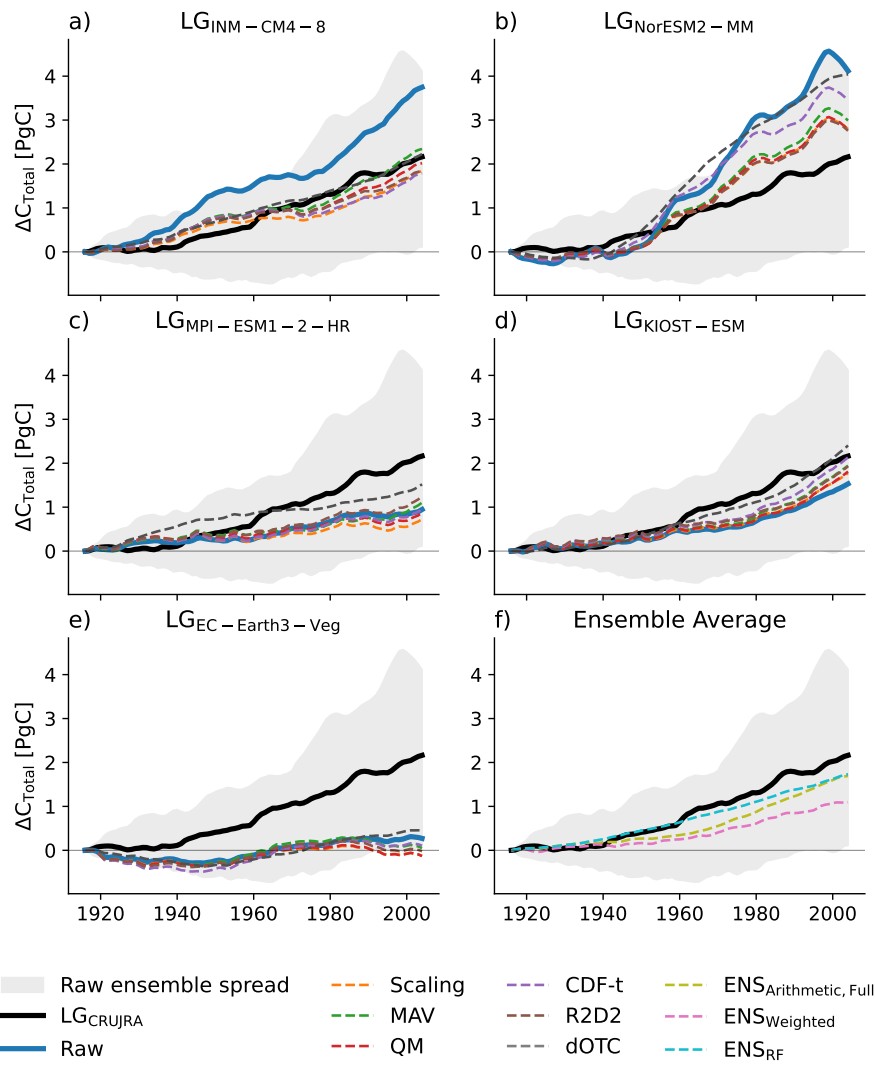

**Figure 4.** 30-year moving average of the change in $C_{Total}$. In each panel, the bold black line is the change in $C_{Total}$ obtained using the CRUJRA reanalysis and the grey shaded area represents the full unconstrained CMIP6 model ensemble. Panel a–e show the $C_{Total}$ change simulated using input from the five bounding models. The colors show the change in $C_{Total}$ based on the different bias correction methods. Panel f shows the change in $C_{Total}$ estimated by the ensemble averaging methods.

Figure 4 shows the change in $C_{Total}$ relative to the 1901–1930 average for the five bounding models (i.e., weakest and highest amount, change and IAV in precipitation over time; see fig. B2 and B1 for the corrected precipitation and temperature forcing). For the LPJ-GUESS runs based on the lowest amount in precipitation and increase in precipitation ($LG_{EC-Earth3-Veg}$ and $LG_{MPI-ESM1-2-HR}$, respectively), none of the bias correction approaches significantly alters the change in $C_{Total}$ so



that the change in $C_{Total}$ remains significantly lower compared to $LG_{CRUJRA}$ (see fig. 4 c and e). In the LPJ-GUESS runs
forced with the highest annual precipitation ($LG_{INM-CM4-8}$) and the strongest increase and highest IAV in precipitation (both
$LG_{NorESM2-MM}$), the bias correction methods generally reduce the simulated change of $C_{Total}$ so that it is closer to the
$LG_{CRUJRA}$ result (see fig. 4 a, b). For $LG_{INM-CM4-8}$, all methods are successful in bias correcting to the reanalysis. For
$LG_{NorESM2-MM}$, four methods approximately halve the difference between the reanalysis and raw runs, with the exception
of CDF-t and dOTC. Figure 4 f shows the impact of different ensemble averaging methods applied to $C_{Total}$. All averaging
methods simulate very similar $\Delta C_{Total}$ in the last 10 years of the model runs whereas the weighted approach is lower by $\sim$0.5
PgC in the first fifty years.





**Figure 5.** Difference between the ensemble averages of $C_{Total}$ and $C_{Total}$ simulated by $LG_{CRUJRA}$. Panel a-c show the arithmetic, weighted, and random forest ensemble average based on the LPJ-GUESS runs using the full CMIP6 ensemble. Panel d-f show the arithmetic ensemble average based on LPJ-GUESS runs using subselections of the CMIP6 ensemble (skilled, independent, and bounding GCMs). Panel g-l show the arithmetic ensemble average based on LPJ-GUESS runs using the bias corrected bounding GCMs following the scaling, MAV, QM, CDF-t, R2D2, and dOTC approach. The noticeable bias across the Tropic of Capricorn results from the assumed bioclimatic limit for C4 grasses.







**Figure 6.** Coefficient of variation (CV) over the ensemble of $C_{Total}$ simulated by LPJ-GUESS. Panel a shows the CV based on the LPJ-GUESS runs using the full CMIP6 ensemble. Panel d-f show the CV based on LPJ-GUESS runs using subselections of the CMIP6 ensemble (skilled, independent, and bounding GCMs). Panel g-l show the CV based on LPJ-GUESS runs using the bias corrected bounding GCMs following the scaling, MAV, QM, CDF-t, R2D2, and dOTC approach. The noticeable CV across the Tropic of Capricorn results from the assumed bioclimatic limit for C4 grasses.




Figure 5 shows the regional details of the relative differences between $C_{Total}$ based on the three ensemble averaging methods (full ensemble; a-c), and different model selection methods (d-e) compared to the reference run $LG_{CRUJRA}$. The arithmetic (see fig. 5a) and weighted average (see fig. 5b) show regional biases that can be both positive (East Central Australia) and negative (Southwest Australia), and along the Tropic of Capricorn. The random forest approach shows small differences in $C_{Total}$ compared to the CRUJRA reanalysis. Figure 5 further supports that using a weighted average or random forest approach yields a more robust ensemble estimate than using the mean of any of the sub-ensembles. Deriving the arithmetic average based on the full ensemble or on a sub-selection based on independent or bounding models (see fig. 5a,e,f) yields very similar results; notably choosing the five most skilled models produces an overall negative bias in the $C_{Total}$ estimate (see fig. 5d).

Correcting the bounding models tends to reduce the bias in the ensemble average of $C_{Total}$ (see fig. 5 g-m). The resulting bias map for individual GCMs can depend on the raw simulation by the GCM to which the bias correction is applied. Each of the bias correction methods leads to similar spatial patterns within the same GCM (see appendix fig. B8).

Figure 6 shows the coefficient of variation (CV) of $C_{Total}$ across the ensemble. Selecting either the full ensemble or making a sub-selection based on skill and independence (see fig. 6a-c), results in a high CV across the Tropic of Capricorn that results from the assumed bioclimatic limit for C4 grasses (similar to fig. 5). Selecting models based on skill (see fig. 6a) reduces the CV compared to the full ensemble while choosing the five bounding models reduces the CV across the Tropic of Capricorn but increases it in most of the other regions. The CV is significantly lower when the climate forcing input is bias corrected for all methods, and the quantile mapping approach overall leads to the lowest values.





**Figure 7.** Boxplots showing the median, 75$^{th}$, and 25$^{th}$ percentiles of foliar projective cover (FPC) for temperate (a) and tropical (b) trees and C3 (c) and C4 (d) grasses. The first five groups are the LPJ-GUESS runs based on the five bounding models LG$_{EC-Earth3-Veg}$, LG$_{INM-CM4-8}$, LG$_{KIOST-ESM}$, LG$_{MPI-ESM1-2-HR}$, and LG$_{NorESM2-MM}$ where blue shows the FPC based on the raw model forcing and orange, green, red, purple, brown and grey show the FPC when LPJ-GUESS is forced with the corrected model forcing following the scaling, MAV, QM, CDF-t, dOTC and R2D2 method, respectively. The yellow, pink and bright blue boxplots on the right hand side of each panel show the different ensemble averaging methods (arithmetic average, weighted average, and random forest, respectively) when the full ensemble is used (group 'Full'). The groups Skill (dashed), Independence (dotted), and Bounding (dashed the other way around) show the results for the arithmetic average when only a sub-selection of models is used (see section 3.1). The dashed lines show the median values of the simulations with the CRUJRA reanalysis, the dotted lines are the 75$^{th}$ and the dash-dotted line the 25$^{th}$ percentiles.



The different patterns in $\Delta C_{\text{Total}}$ for the bounding model runs imply that the underlying vegetation composition might vary
with the climate forcing and the bias correction methods applied. Indeed, studies have suggested that the sensitivity to climate
forcing is generally larger on regional and PFT-scales (Wu et al., 2017). To examine the impact of bias correction on vegeta-
tion composition we examine the FPC of four different vegetation groups (temperate and tropical trees, C3 and C4 grasses)
for the five bounding models and different ensemble averages (see fig. 7). For temperate trees, most raw models simulate a
higher median compared to the FPC based on $LG_{\text{CRUJRA}}$ (except for MPI-ESM1-2-HR; see fig. 7 a) and the variability in
simulated FPC depends strongly on the GCM used to drive LPJ-GUESS. For the LPJ-GUESS runs based on the wettest GCM
($LG_{\text{INM-CM4-8}}$ and the one based on the strongest increase in precipitation ($LG_{\text{NorESM2-MM}}$), the median falls outside the
$LG_{\text{CRUJRA}}$ interquartile range and the 75$^{\text{th}}$ percentile of both models is more than double ($LG_{\text{MPI-ESM1-2-HR}}$) or triple
($LG_{\text{INM-CM4-8}}$) of what the $LG_{\text{CRUJRA}}$ run suggests. For all models, correcting the GCM forcing brings the simulated FPC
much closer together. The arithmetic and weighted ensemble average result in a higher median and 25$^{\text{th}}$ and 75$^{\text{th}}$ percentile
compared to the $LG_{\text{CRUJRA}}$ run. The median of random forest is close to the $LG_{\text{CRUJRA}}$ median. However, 75$^{\text{th}}$ is signifi-
cantly lower compared to that of $LG_{\text{CRUJRA}}$ and the variability for the random forest approach is overall lower compared to
$LG_{\text{CRUJRA}}$. Only choosing skilled models reduces the median of the arithmetic ensemble average, leading to better agreement
with the $LG_{\text{CRUJRA}}$ reanalysis but the variability is lower. The other selection methods produce similar values for the median
compared to the full ensemble result with a larger spread.

For the tropical trees (see fig. 7 b), most models simulate medians and interquartile ranges similar to that based on the
$LG_{\text{CRUJRA}}$ reanalysis. In contrast, the FPC based on wettest GCM ($LG_{\text{INM-CM4-8}}$) shows a significantly higher median and
75$^{\text{th}}$ percentile (the latter about four times higher compared to $LG_{\text{CRUJRA}}$). All bias correction methods decrease the median so
that it is within the $LG_{\text{CRUJRA}}$ interquartile range (IQR). The MAV approach however still leads to a too high 75$^{\text{th}}$ percentile.
The weighted ensemble average shows the distribution that is the most similar compared to the $LG_{\text{CRUJRA}}$ FPC. Calculating
the arithmetic average based on the full ensemble yields a similar result, however the random forest approach median almost
drops out of the $LG_{\text{CRUJRA}}$ IQR. The arithmetic approach based on the independent GCMs produce the best match compared
to $LG_{\text{CRUJRA}}$.

In contrast to the two tree groups, the median C3 grass FPC based on the CMIP6 forcing tends to be lower than that based on
$LG_{\text{CRUJRA}}$ (see fig. 7). The C4 grasses show a mixed response to the raw CMIP6 forcing. The LPJ-GUESS runs based on the
wettest model and the one with the strongest increase in precipitation ($LG_{\text{INM-CM4-8}}$ and $LG_{\text{NorESM2-MM}}$ simulate a higher
median FPC compared to the $LG_{\text{CRUJRA}}$ while the runs based on the driest model and the model with the lowest increase
in precipitation ($LG_{\text{MPI-ESM1-2-HR}}$ and $LG_{\text{EC-Earth3-Veg}}$) are lower. Especially the $LG_{\text{MPI-ESM1-2-HR}}$ run shows large
variation in simulated C4 grass FPC depending on the correction method. For $LG_{\text{INM-CM4-8}}$, the three approaches based on
quantile mapping (QM, CDF-t and dOTC) lower the median closer to the $LG_{\text{CRUJRA}}$ median. For the wet model, all approaches
lead to significant improvement. None of the arithmetic or weighted ensemble averages in FPC match the $LG_{\text{CRUJRA}}$ median,
and mostly are below the lower quartile of $LG_{\text{CRUJRA}}$.

Overall, the analysis of FPC highlights important implications for bias correction. The results show that LPJ-GUESS re-
sponds very differently to the various bias correction methods because the change in the GCM forcing alter the competitive



interactions between vegetation types. Importantly, although the spatial maps show similar agreement in $C_{Total}$ between cor-
rection methods, the change in FPC implies that the resulting change in carbon is simulated by difference underlying vegetation
compositions. We therefore further examine the seasonal cycle of GPP of C4 grasses in the following as the change was the
most different after bias correction.

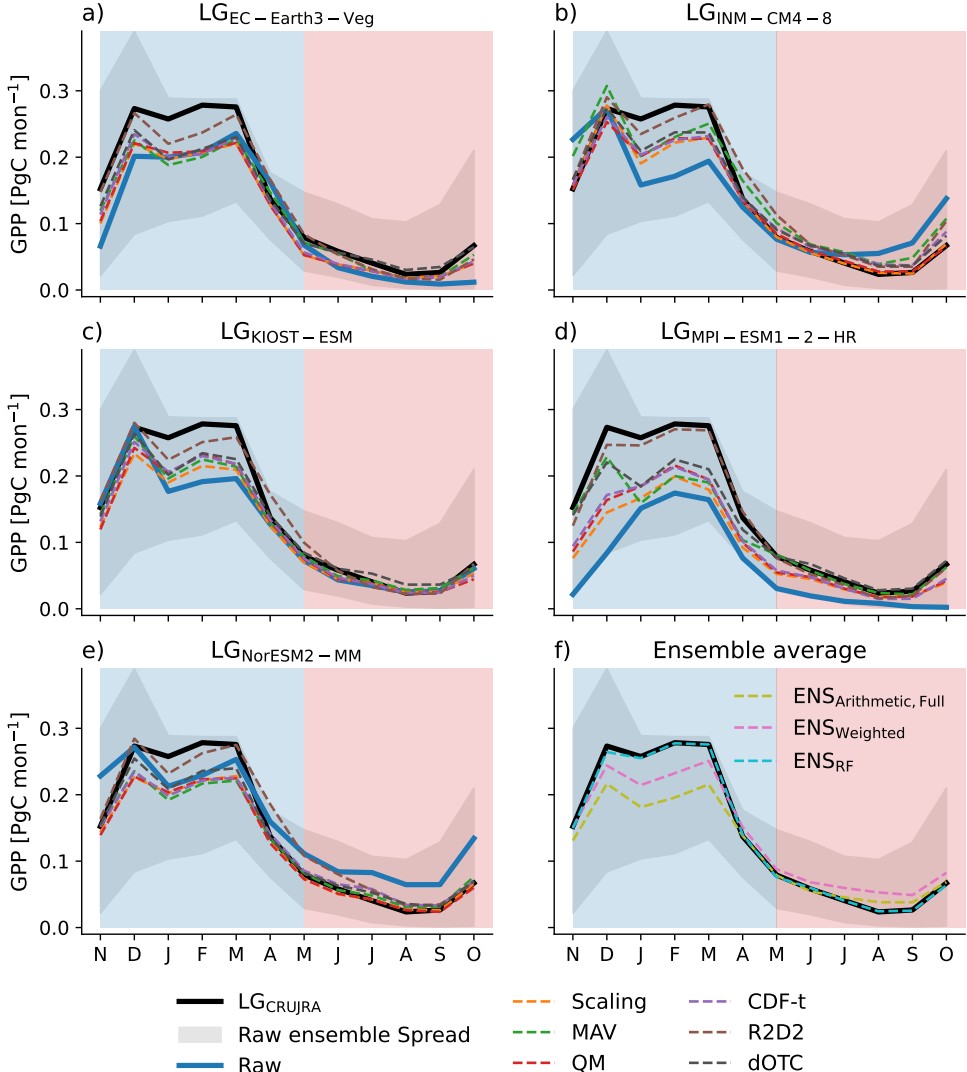

**Figure 8.** Seasonal cycle of gross primary productivity for C4 grasses. The different panels show the seasonality when LPJ-GUESS is
forced with the bounding five bounding models (a-e). The different colors show the unconstrained model climate forcing (blue), or after bias
correcting the data following the scaling (orange), the mean and variance (green), the quantile mapping (red), the CDF-t (purple), the dOTC
(brown) and the R2D2 (grey) method. The black lines represent the reanalysis simulations with CRUJRA and the grey shading shows the full
CMIP6 ensemble spread. The blue shaded area indicate the wet season (November–April) and the red area the dry season (May–October).





Figure 8 shows the seasonal GPP for C4 grasses. All simulations, including $LG_{CRUJRA}$, simulate peak productivity in the wet season and minimum productivity in the dry season (see fig. 8 a). Through December to March, the maximum GPP during
the wet season is lower compared to the reanalysis results but is closer to the reanalysis simulations in the dry season. As a result, the bias correction methods achieve similar $C_{Total}$ values (see fig 3) predominantly through reducing biases during the dry season and introducing an underestimation bias in the wet season. For $LG_{MPI-ESM2-2-HR}$, the raw climate forcing does not generate the right magnitude and timing of peak GPP. When corrected with the two multivariate approaches, both become more similar to the $LG_{CRUJRA}$ runs. For $LG_{INM-CM4-8}$ and $LG_{MPI-ESM1-2-HR}$, all bias correction methods increase GPP
from December to March, while for $LG_{KIOST-ESM}$, only the two multivariate approaches achieve a change closer to the $LG_{CRUJRA}$ runs in the wet season GPP. When the NorESM2-MM climate forcing is corrected, the magnitude is even lower than when the raw climate forcing is used. Figure 8 f also shows the impact of the different ensemble averaging approaches. Applying the random forest approach leads to near identical result to the $LG_{CRUJRA}$ simulation. Both the weighted and arithmetic ensemble average result in a lower peak in GPP in the wet season, where the arithmetic average is lower than both
the random forest result and the weighted average.

## 5    Discussion

In this study, we explored the impact of climate model uncertainty on the regional carbon cycle over Australia and the sensitivity of the carbon cycle to different approaches to correcting climate forcing biases. We found that, uncorrected, the continental-scale climate projections over Australia were associated with large uncertainties. The difference between the hottest and coldest
model is very large; 3.4°C higher than the observed historical warming over the continent (1.4°C; IPCC, 2021), and local differences can be even larger. Similarly, average precipitation ranges between 254 and 858 mm yr$^{-1}$, and the IAV ranges from 55-183 mm yr$^{-1}$. The differences on both timescales have a large impact on predicted vegetation, especially across a water-limited continent such as Australia. Our finding that the simulation of Australia's carbon cycle is particular sensitive to the choice of climate forcing is consistent with previous studies (e.g. Ahlström et al., 2012; Ahlström et al., 2015; Ahlström
et al., 2017). The uncertainty in the CMIP6 forcing translates into a significant variability in the simulated carbon cycle in LPJ-GUESS, for example the average values for $C_{Total}$ vary between 28.6 PgC and 75.1 PgC, and the IAV in NBP was between 0.3 and 1.1 PgC. While Australia is not the largest contributor to the global carbon sink on centennial timescales, the continents' total carbon storage is still significant. On shorter timescales, the IAV in NBP is important for the both historical and future estimates of atmospheric growth rate since several studies (e.g. Poulter et al., 2014; Ahlström et al., 2015) have found that
Australia can be a major contributor to the global net carbon sink in wet years. It is therefore important to reduce the uncertainty in carbon cycle projections over Australia, first to improve estimates of future carbon sinks, second to help constrain future atmospheric growth rates and third, because the improved understanding will ultimately enable better predictions of vegetation responses to climate change over Australia. We explored three approaches to reduce biases and ensemble uncertainty and discuss each in turn below.



## 5.1 Sensitivity to bias correction methods

We tested six different methods for bias correcting the CMIP model forcing driving LPJ-GUESS. Four methods incorporate univariate approaches (each climate variable is corrected independently), and two employ multivariate approaches (inter-variable relationships are accounted for). We found that all bias correction methods reduce the average bias of $C_{Total}$ to that of the reference run for the five individual models. When deriving an arithmetic ensemble average of the raw and bias corrected results, the values for the ensemble averages are relatively similar. Correcting the climate forcing significantly reduces the spread amongst the ensemble members compared to the raw model forcing. We further explored regional differences in the $C_{Total}$ bias compared to our reference run, and found that all bias corrections methods reduce the magnitude of the bias. The spatial patterns in bias were consistent across the bias correction methods, implying that the relative spatial distribution of $C_{Total}$ remains similar.

In contrast to the average $C_{Total}$ results, bias correcting the forcing CMIP models does not necessarily lead to better results for other variables simulated by LPJ-GUESS. The different bias correction approaches did not necessarily lead to improved simulations of the change in $C_{Total}$. The arithmetic average across all five bounding models is relatively close to that of the reference run, and the upper boundary of the model spread was reduced when bias correction methods were applied. However, the lower boundary was almost the same or slightly worse than before (EC-Earth3-Veg). The different biases and magnitudes in $C_{Total}$ reflect that the underlying vegetation composition may vary depending on the CMIP6 ensemble member used to run LPJ-GUESS, and the bias correction method.

The foliar projective cover gives an indication of the fidelity of vegetation cover. We found that temperate trees and C4 grasses in particular can vary strongly in dominance and relative cover depending on the GCM used as the input forcing and bias correction applied. For example, the distribution of C4 grasses based on the MPI-ESM1-2-HR climate forcing does not overlap with any of the other unconstrained models. Only the two multivariate approaches adjust the distribution so that it is more comparable to the reference dataset and the other ensemble members. This implies that both the model selection as well as the bias correction method can lead to small but potentially important differences in composition of vegetation distributions across the landscape. Models that show large differences in the vegetation distribution are also sensitive to the bias correction for seasonal GPP. For the two models with the strongest divergence in C4 grass distribution, all bias correction methods improve the seasonal productivity. However, correcting the climate forcing also led to a lower skill in predicting seasonal GPP for one model ($LG_{NorESM2-MM}$). We also found the foliar projective cover, especially that of C4 grasses, showed a strong sensitivity to the bias correction method chosen for some models (e.g. MPI-ESM1-2-HR). However, the spatial patterns in average bias of $C_{Total}$ remain relatively consistent across all bias correction methods tested and show some similarity to that of the raw model forcing ($LG_{EC-Earth3-Veg}$, $LG_{KIOST-ESM}$ and $LG_{NorESM2-MM}$). This outcome may emerge as we corrected each grid cell independently. When François et al. (2020) correct their climate variables taking into account spatial properties, both methods tested here improved the results for small regional scales. Given the heterogeneity of climate and large area of the Australian continent, we did not attempt correcting the spatial scales given limitations in computation time but this would be worth exploring in future work.





In summary, within a framework of testing bias correction methods on the five models spanning the CMIP6 model spread,
we found that the bias correction methods successfully reduced the bias to the reference dataset for averages over time and
space ($C_{Total}$) but show limited impact for other temporal properties (such as the change over time; e.g. Hagemann et al.,
2011; Maurer and Pierce, 2014; Cannon et al., 2015; François et al., 2020). For example, Hagemann et al. (2011) found that
bias correction does not necessarily lead to a more realistic climate change signal. In a different study focusing on precipitation,
Maurer and Pierce (2014) demonstrated that long-term changes in simulated precipitation can artificially deteriorate following
quantile mapping. Further, Cannon et al. (2015) find that quantile mapping approaches can inflate relative trends in precipitation
extremes projected by GCMs. The lack of skill in correcting temporal properties was also demonstrated for multivariate bias
correction approaches (François et al., 2020). Using single models or even a subset of the ensemble may therefore not inform
trends and processes on short timescales for studies exploring the future carbon cycle. Despite the demonstrated limited impact
of bias correction on temporal and spatial scales, correcting the driving forcing is still preferable to using raw climate forcing.
DGVMs largely rely on bioclimatic limits that define where specific types of vegetation can grow. Relying on a biased climate
forcing dataset might therefore result in a misrepresentation of the vegetation. Indeed, we found strong differences in the foliar
projective cover of different vegetation groups. This mismatch in vegetation composition that can result from threshold-defined
boundaries is likely to lead to diverging carbon and water cycle responses to the climate, which might be even more pronounced
in areas with higher vegetation carbon mass than Australia. Future studies could further explore options to improve temporal
features in climate variables. Robin and Vrac (2021), for example, include time as an additional variable for their multivariate
bias correction which may be a promising avenue for future research.

Climate change impact studies need to be aware of the limitations of bias correction methods. As we have shown, bias
correction cannot solve fundamental deficiencies in GCMs (Maraun et al., 2017). A possible flaw in applying univariate bias
correction methods on a set of climate variables needed to force a dynamic vegetation model is a resulting inconsistency within
the climate forcing. While all bias correction methods improve the averages of $C_{Total}$, importantly, based on our findings
it is not clear that one method systematically outperforms any other. This may be because the carbon cycle in Australia is
mostly driven by precipitation, and for vegetation limited by both temperature and precipitation, multivariate approaches may
outperform univariate approaches more distinctly (Zscheischler et al., 2019). While the ensemble average is mostly insensitive
to choice of raw or corrected data, the spread between the outlier models is significantly reduced by any of the correction
methods (especially the quantile mapping approaches and the multivariate dOTC method). Other temporal properties, such
as the change over time, are not necessarily improved or can even deteriorate compared to the raw climate forcing, such as
the trend, interannual variability or extreme events. Researchers should be especially cautious when they rely on a small sub-
sample or even single models for their impact study, given different GCMs can react differently to the same bias correction
method (e.g. for $LG_{INM-CM4-8}$, the magnitude in bias is reduced while for $LG_{NorESM2-MM}$ the sign in bias can change
depending on correction method applied).



## 5.2 Sensitivity to ensemble averaging methods and model selection methods

We also tested the commonly used arithmetic ensemble average, a weighted averaging approach following Bishop and Abramowitz (2013), and a random forest regression approach. We found that the weighted average and the random forest approach outperform the arithmetic ensemble average for average $C_{Total}$, and seasonal GPP with results very similar to the reference dataset.

The random forest approach produces a small error magnitude when spatial dimensions are explored (see fig. 5) while for the arithmetic and weighted ensemble average, systematic biases persist. While the FPC of tropical trees and C3 grasses seems to be broadly captured by all averaging methods, C4 grasses shows a strong bias where only the random forest approach achieves a median value within the IQR of the $LG_{CRUJRA}$ run. As shown in previous studies (e.g. Bishop and Abramowitz, 2013; Knutti et al., 2017; Abramowitz et al., 2019; Merrifield et al., 2020) there is benefit to avoiding the use of the arithmetic ensemble

averaging method for impact studies. An additional caveat of the arithmetic ensemble average is the sensitivity to the model selection. The ensemble average somewhat depends on the models it is derived from. Counter intuitively, choosing the models that show high skill in simulating precipitation, led to the worst results in most cases (a result similar to Herger et al., 2018).

## 5.3 General caveats

All methods explored in this study rely on the general assumption that the reanalyses used to describe the historical time period

are accurate and that the methods employed apply equally to the past and the future. It seems reasonable to argue that methods that fail to constrain models in the historical period are unlikely to work well for future periods. Unfortunately, the converse that methods that work well in the historical period will necessarily work well in the future is not always true. Shifts in atmospheric circulation, emergence of novel climates or the triggering of ecosystem tipping points might alter land-atmosphere feedbacks that lead to changes in the climate such that methods that are reliable in the historical period cease to be reliable in the future.

A possible caveat in our study set up is the design of the ensemble subsets. We selected all models based on the simulated precipitation based on the assumption that precipitation is the most important driver of Australia's carbon cycle. However, temperature and perhaps the extremes of temperature may also be an important constraint for vegetation distribution (especially in LPJ-GUESS where vegetation grows within pre-defined bioclimatic limits that are based on temperature like the boundary between C3 and C4 grasses). However, when we repeated the analysis using the raw temperature and incoming shortwave

radiation forcing and bias corrected precipitation data, the results were almost identical compared to the runs where all climate variables were corrected, confirming that precipitation drives the carbon cycle response within this framework. Further, for simulating vegetation the skill of the variables may be important on multiple timescales. We attempt to account for this in the model selection methods by applying the respective metrics on monthly and annual timescales. In addition, five models for all selection methods may seem like a small subset. However, earlier studies (e.g. Pierce et al., 2009) found that the multi-model

ensemble mean tends to converge towards a similar value after including five models. We therefore conclude that five models was a sufficient number in our testing framework.

We further chose a relatively short calibration time period (1989–2010) to allow sensitivity tests with multiple reanalysis datasets. While these 22 years may not cover decadal variability, we assume it is sufficient to account for interannual variability





such as the El Niño Southern Oscillation, the Indian Ocean Dipole, and Southern Annual Mode which have been shown to be
important influences on the Australian carbon cycle (e.g. Cleverly et al., 2016).

Other areas of uncertainty may include the sensitivity of the methods to the reference dataset. Several studies have discussed
that both bias correction methods (e.g. Iizumi et al., 2017; Famien et al., 2018; Casanueva et al., 2020), and weighted ensemble
averaging methods (e.g. Merrifield et al., 2020) depend on the observation dataset they are calibrated on. Casanueva et al.
(2020) demonstrate that precipitation in particular is sensitive to the choice of reference dataset. We therefore repeated the bias
correction and chose ERA5 as a second dataset. We found high correlation coefficients between LPJ-GUESS runs that are based
on GCMs corrected to CRUJRA and LPJ-GUESS runs that were based on GCMs corrected to ERA5 for $C_{Total}$ (0.96–0.98; see
appendix fig. B3-B7). We conclude that our results were robust to the choice of reference dataset. Another concern frequently
discussed is impact of the mismatch in spatial resolution (high resolution reanalysis product vs. low resolution GCM output).
A solution to reduce the mismatch in spatial resolution might be to use dynamically downscaled datasets, such as CORDEX.
However, Casanueva et al. (2020) find the impact of the horizontal resolution on the bias correction results to be small in
comparison to the impact of bias correction method. Given dynamically downscaled products were only available for older
CMIP generations (CORDEX is based on CMIP5, NarCLIM on CMIP3) or contained a small subset of GCMs only (ISIMIP),
and we expected the uncertainty associated with the spatial mismatch to be small, we chose the state-of-the-art CMIP6 GCM
output.

Lastly, we chose to correct daily climate data for the main analysis. However, correcting monthly data may be statistically
more robust. We additionally tested the importance of timescales, i.e., we bias corrected the GCMs on both daily and monthly
timescales before forcing LPJ-GUESS with them. $C_{Total}$ simulated by LPJ-GUESS driven by daily and monthly corrected
GCM output was strongly correlated (0.92–0.99; see fig. B3-B7).

### 5.4 Implications

Based on our findings, we conclude that decisions in regard to model selection, bias correction of GCM output, and ensemble
averaging methods, may alter future projections of ecosystem studies, especially the uncertainty estimates. Selecting a subset of
models to reduce computation time is common, but sensitive to the criterion chosen for both arithmetic average and uncertainty
estimate. While choosing GCMs based on how well they represent the historical climate may seem intuitive, we find that the
arithmetic average based on a subset representing only independent models or models that define the full ensemble spread
reduces the bias compared to our reference run. Conversely, a subset of only skilled models reduces the ensemble uncertainty.
However, this reduction in uncertainty may stem from the wrong biophysical reasons, and a sub-selection of skilled models
might not truly represent all plausible GCM outputs.

We further demonstrate that correcting GCM output can significantly alter Australia's carbon cycle projections. Bias correc-
tions however only reduce the biases in relatively steady vegetation variables, such as the longer-term carbon states. Averaged
over the continent, we find that LPJ-GUESS forced with individual corrected GCM output can be sensitive to the bias cor-
rection method but the arithmetic ensemble averages were found to be insensitive. Some bias correction methods did reduce
the ensemble uncertainty more than others (e.g. Scaling vs. dOTC). On smaller scales, i.e., exploring regional differences or



on PFT level, the choice of bias correction method can have a big influence on species distribution and magnitude in fluxes. Correcting biases may also lead to different outcomes relying on thresholds of absolute values when applied to individual
GCMs, such as for climate threshold studies exploring tipping points.

Importantly, bias correction methods do not correct temporal (such as IAV or trend) and spatial properties, unless the methods are specifically designed and set-up to do so. We found that using corrected GCM output can even increase the distance in change compared to our reference dataset. Future studies of ecosystem/carbon cycle impacts based on GCM climate forcing should therefore carefully choose a subset of models that is representative of the ensemble uncertainty, and do not rely on using
a single GCM.

*Code and data availability.*   The CMIP6 output used in this study is available via the Earth System Grid Federation (ESGF). The CRUJRA reanalysis dataset is accessible via https://catalogue.ceda.ac.uk/uuid/7f785c0e80aa4df2b39d068ce7351bbb (last access: March 2021). The analysis code can be found on https://github.com/lteckentrup/CMIP6_australia.





## Appendix A

**A1**

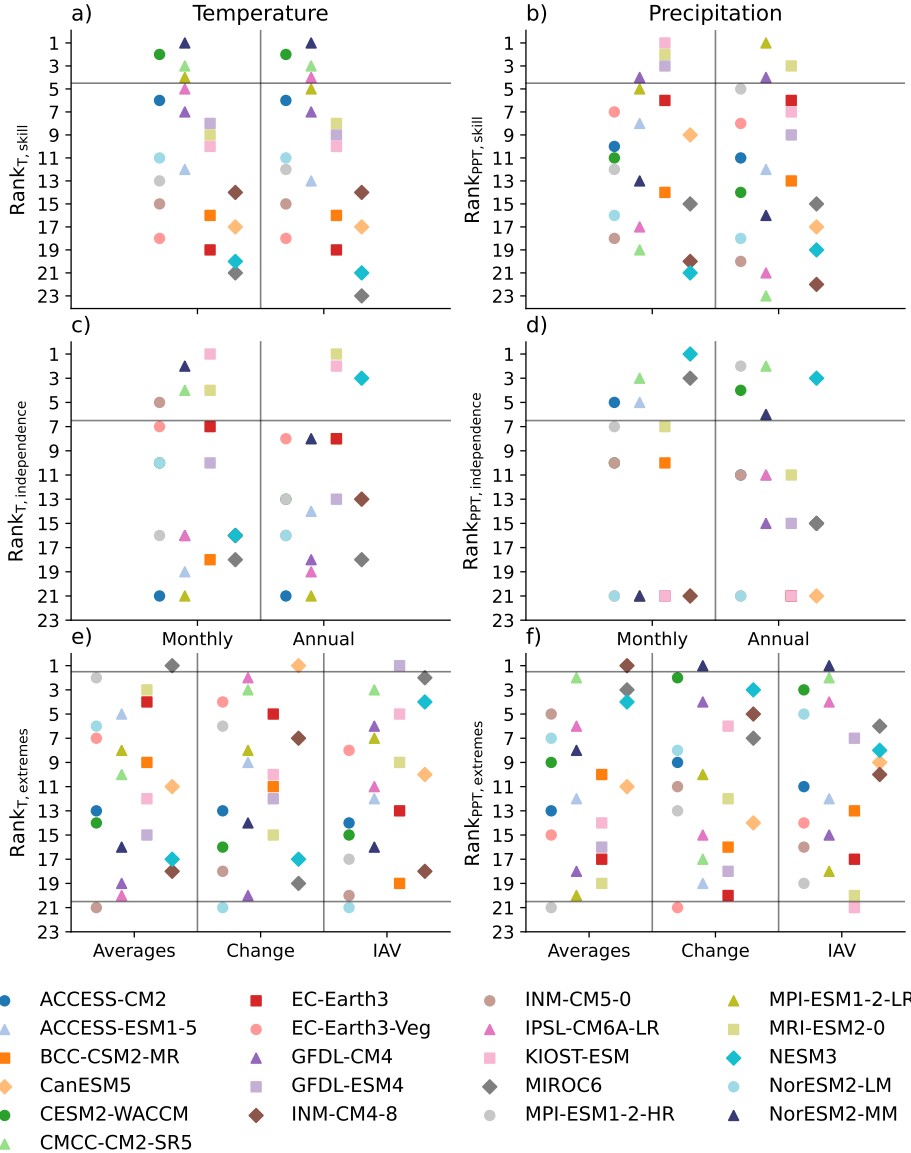

**Figure A1.** Ranks derived for CMIP6 GCM subselection. Panel a and b show the rank according to the skill of each GCM in simulating temperature (a) and precipitation (b) on monthly and annual timescales (compare tab. 2 and section 3.1.1). Panel c and d show the independence rank of each GCM for temperature (c) and precipitation (d) on monthly and annual timescales (compare section 3.1.2). Lastly, panel e and f show the GCMs defining the ensemble spread, i.e. the GCM simulating the highest and lowest total amount in precipitation ('Averages'), change in precipitation ('Change'), and interannual variability ('IAV'; compare section 3.1.3).

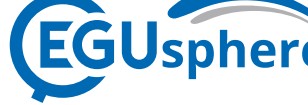

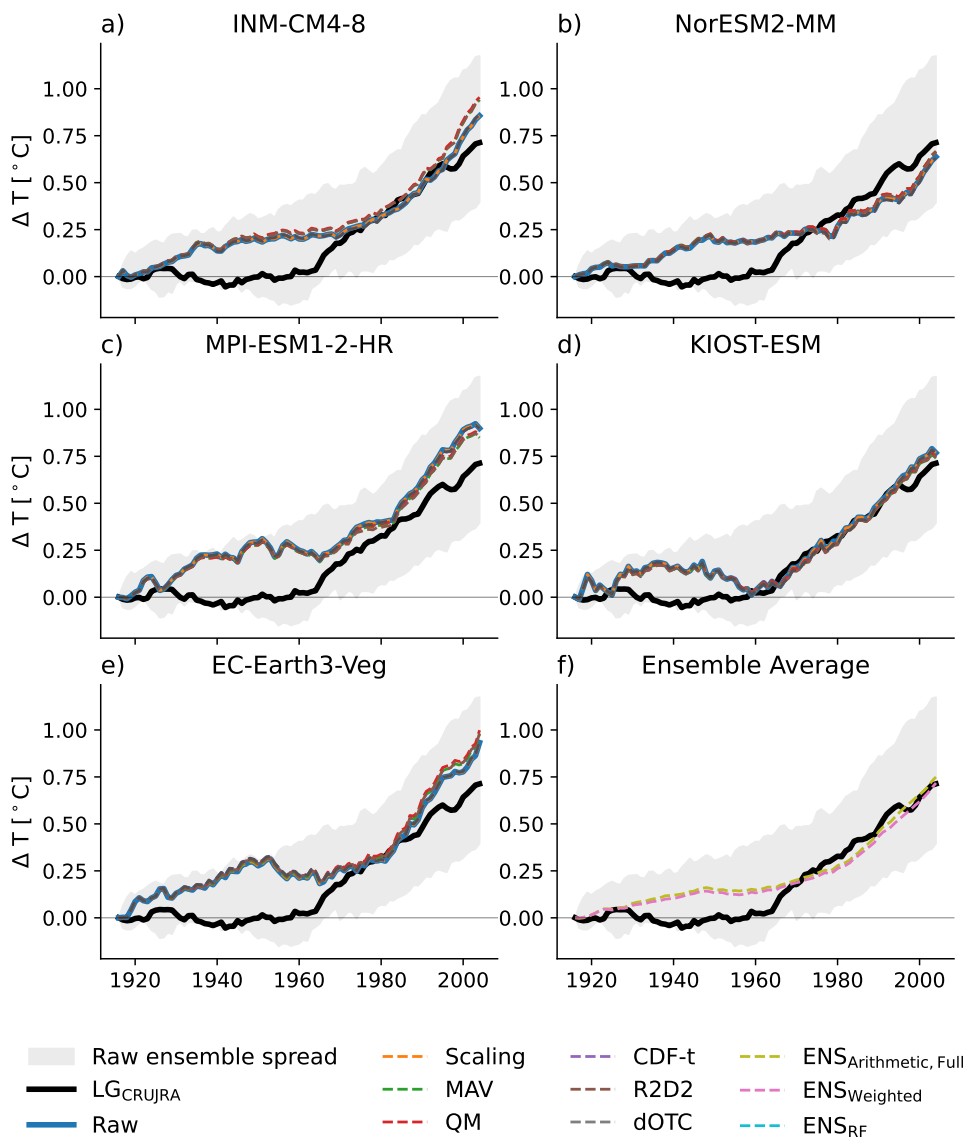

**Figure B1.** 30 year moving average of the change in temperature (T). In each panel, the bold black line is the change in T based on the CRUJRA reanalysis and the grey shaded area represents the full unconstrained CMIP6 model ensemble. Panel a–e show the T change based on the five bounding models. The colors show the change in T based on the different bias correction methods. Panel f shows the change in T estimated by the ensemble averaging methods.



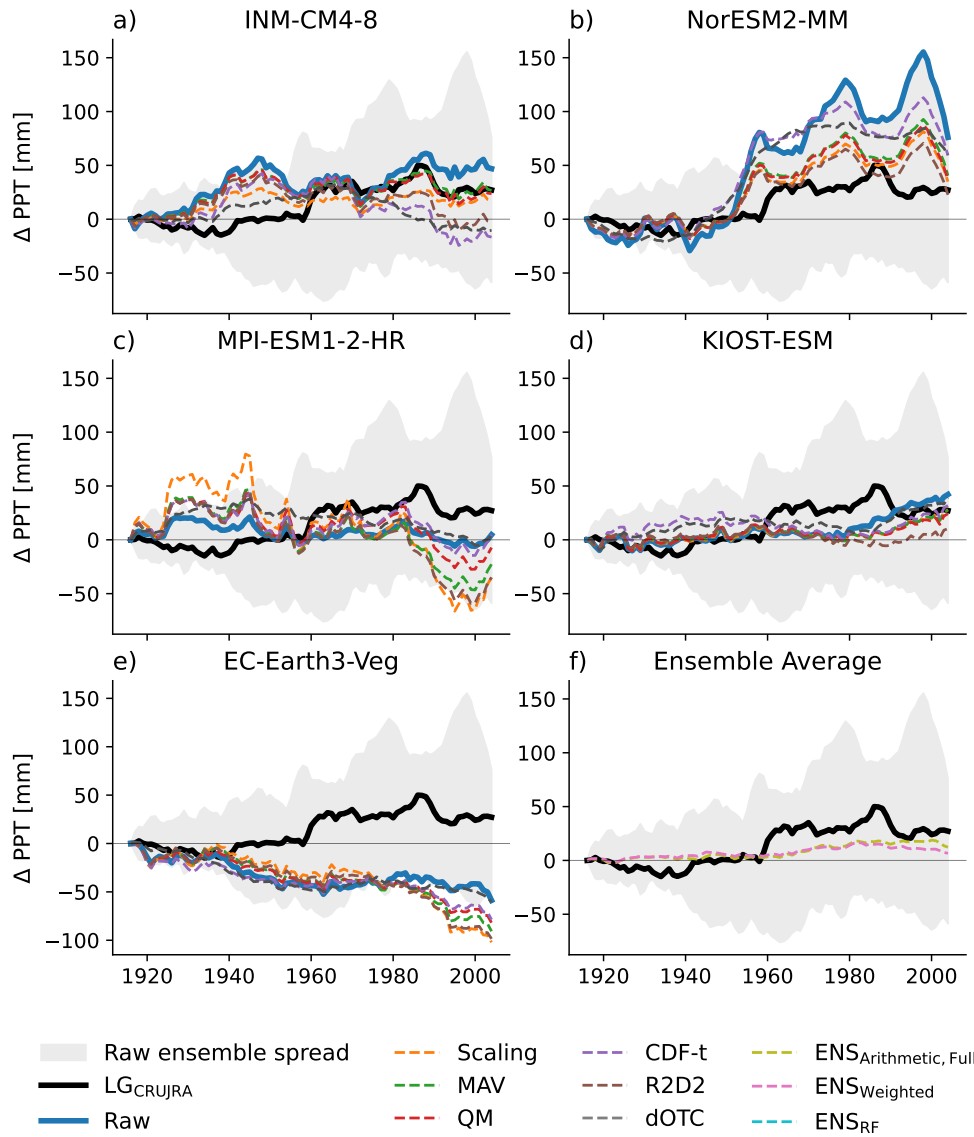

**Figure B2.** 30 year moving average of the change in precipitation (PPT). In each panel, the bold black line is the change in PPT based on the CRUJRA reanalysis and the grey shaded area represents the full unconstrained CMIP6 model ensemble. Panel a–e show the PPT change based on the five bounding models. The colors show the change in PPT based on the different bias correction methods. Panel f shows the change in PPT estimated by the ensemble averaging methods.



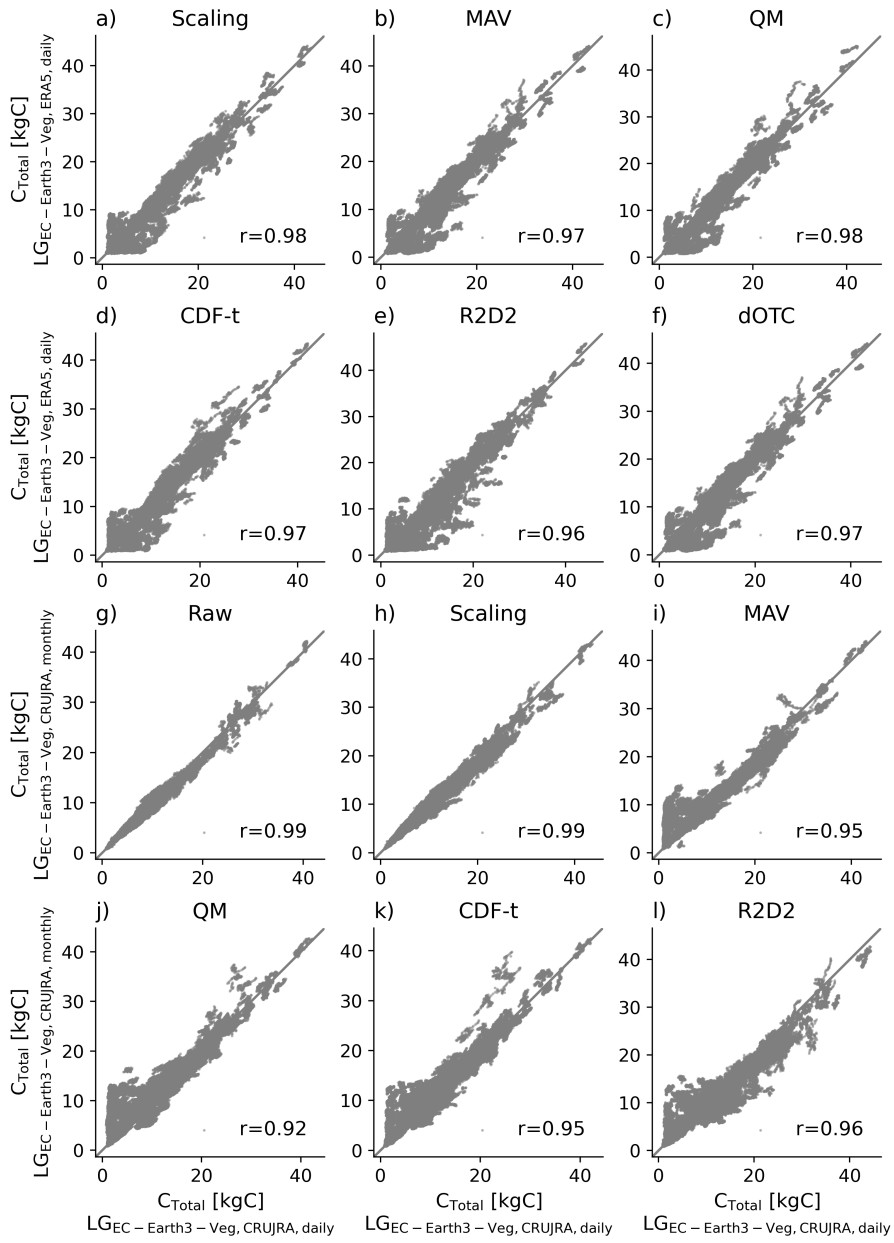

**Figure B3.** Sensitivity to reference dataset (a-e) and to timescale (f-k). Panel a-e show $C_{Total}$ simulated by LPJ-GUESS forced with the EC-Earth3-Veg climate forcing corrected on daily timesteps using the CRUJRA reanalysis as a reference dataset on the x-axis, and LPJ-GUESS forced with the EC-Earth3-Veg climate forcing corrected on daily timesteps using the ERA5 reanalysis as a reference dataset on the y-axis. Panel f-k show $C_{Total}$ simulated by LPJ-GUESS forced with the EC-Earth3-Veg climate forcing corrected on daily timesteps using the CRUJRA reanalysis as a reference dataset on the x-axis, and $C_{Total}$ simulated by LPJ-GUESS forced with the EC-Earth3-Veg climate forcing corrected on monthly timesteps using the CRUJRA reanalysis as a reference dataset on the y-axis. Each panel also contains the pearson correlation coefficient ('r').



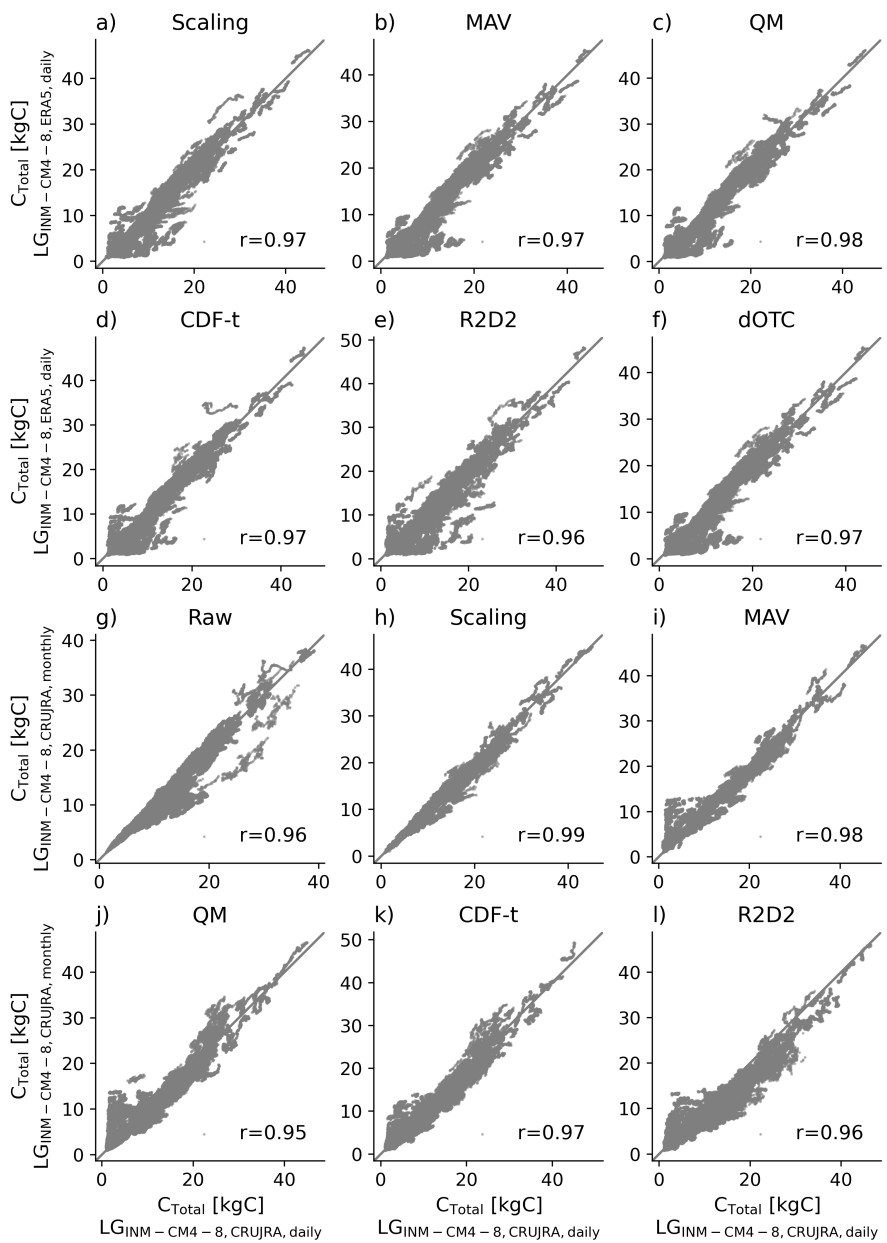

**Figure B4.** Sensitivity to reference dataset (a-e) and to timescale (f-k). Panel a-e show $C_{Total}$ simulated by LPJ-GUESS forced with the INM-CM4-8 climate forcing corrected on daily timesteps using the CRUJRA reanalysis as a reference dataset on the x-axis, and $C_{Total}$ simulated by LPJ-GUESS forced with the INM-CM4-8 climate forcing corrected on daily timesteps using the ERA5 reanalysis as a reference dataset on the y-axis. Panel f-k show $C_{Total}$ simulated by LPJ-GUESS forced with the INM-CM4-8 climate forcing corrected on daily timesteps using the CRUJRA reanalysis as a reference dataset on the x-axis, and $C_{Total}$ simulated by LPJ-GUESS forced with the INM-CM4-8 climate forcing corrected on monthly timesteps using the CRUJRA reanalysis as a reference dataset on the y-axis. Each panel also contains the pearson correlation coefficient ('r).




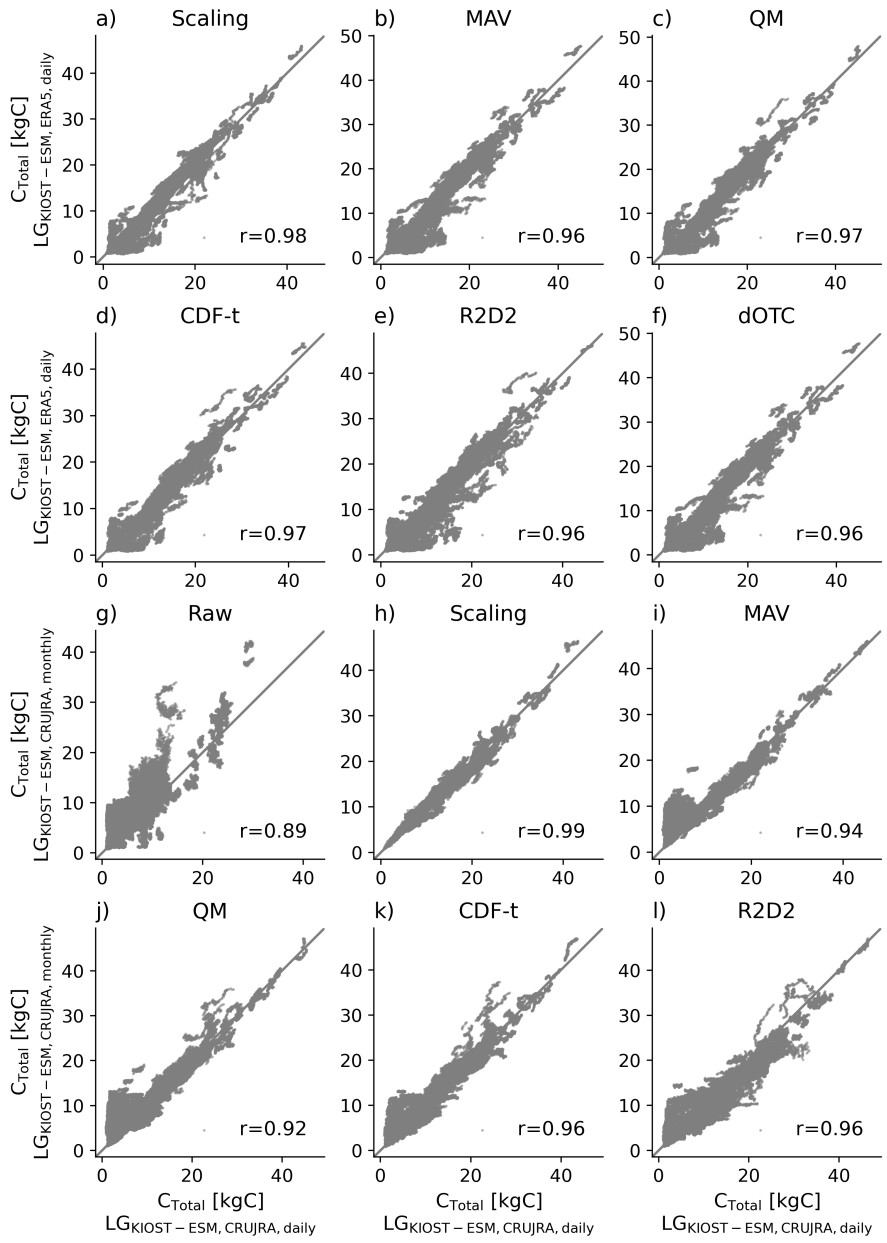

**Figure B5.** Sensitivity to reference dataset (a-e) and to timescale (f-k). Panel a-e show $C_{Total}$ simulated by LPJ-GUESS forced with the KIOST-ESM climate forcing corrected on daily timesteps using the CRUJRA reanalysis as a reference dataset on the x-axis, and $C_{Total}$ simulated by LPJ-GUESS forced with the KIOST-ESM climate forcing corrected on daily timesteps using the ERA5 reanalysis as a reference dataset on the y-axis. Panel f-k show $C_{Total}$ simulated by LPJ-GUESS forced with the KIOST-ESM climate forcing corrected on daily timesteps using the CRUJRA reanalysis as a reference dataset on the x-axis, and $C_{Total}$ simulated by LPJ-GUESS forced with the KIOST-ESM climate forcing corrected on monthly timesteps using the CRUJRA reanalysis as a reference dataset on the y-axis. Each panel also contains the pearson correlation coefficient ('r').





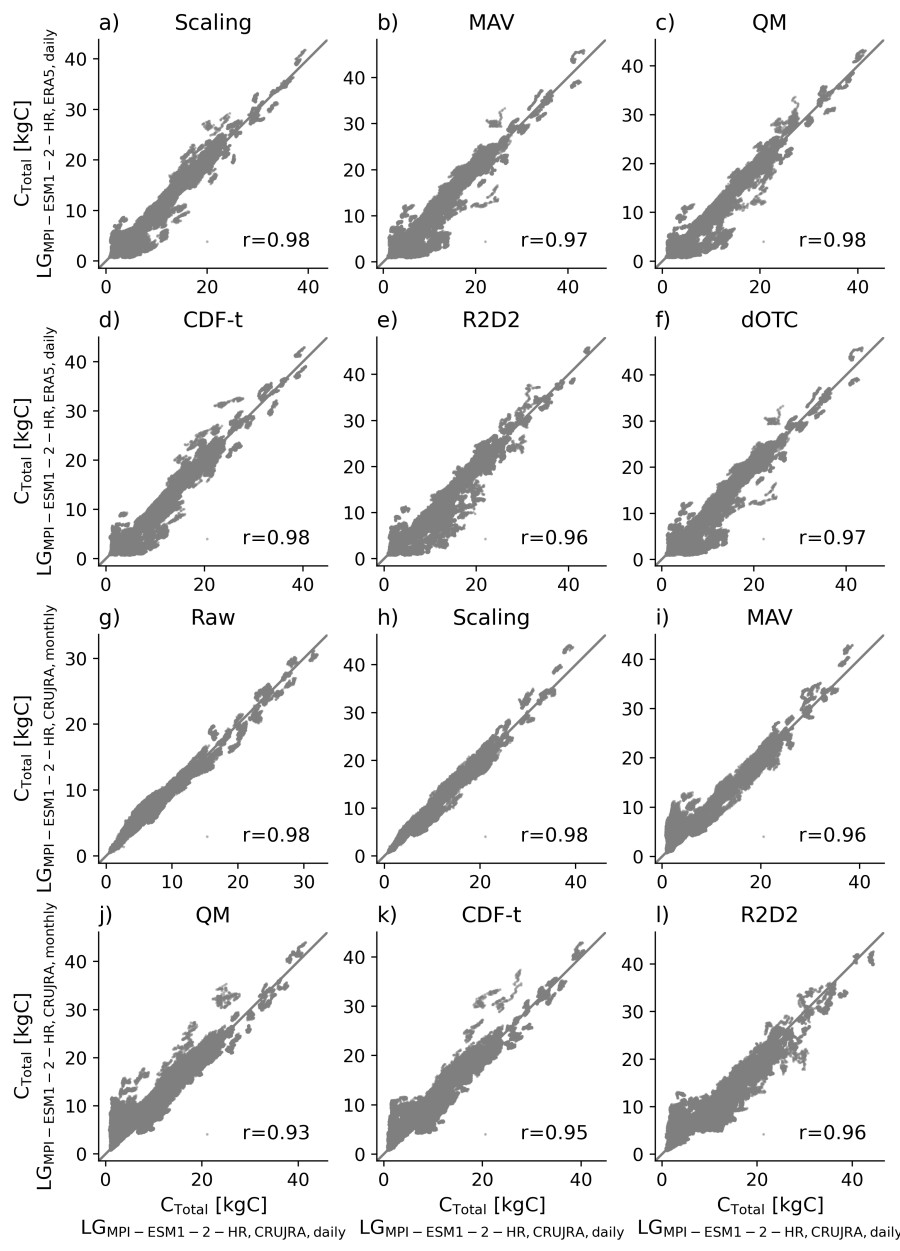

**Figure B6.** Sensitivity to reference dataset (a-e) and to timescale (f-k). Panel a-e show $C_{Total}$ simulated by LPJ-GUESS forced with the MPI-ESM1-2-HR climate forcing corrected on daily timesteps using the CRUJRA reanalysis as a reference dataset on the x-axis, and $C_{Total}$ simulated by LPJ-GUESS forced with the MPI-ESM1-2-HR climate forcing corrected on daily timesteps using the ERA5 reanalysis as a reference dataset on the y-axis. Panel f-k show $C_{Total}$ simulated by LPJ-GUESS forced with the MPI-ESM1-2-HR climate forcing corrected on daily timesteps using the CRUJRA reanalysis as a reference dataset on the x-axis, and $C_{Total}$ simulated by LPJ-GUESS forced with the MPI-ESM1-2-HR climate forcing corrected on monthly timesteps using the CRUJRA reanalysis as a reference dataset on the y-axis. Each panel also contains the Pearson correlation coefficient ('r').



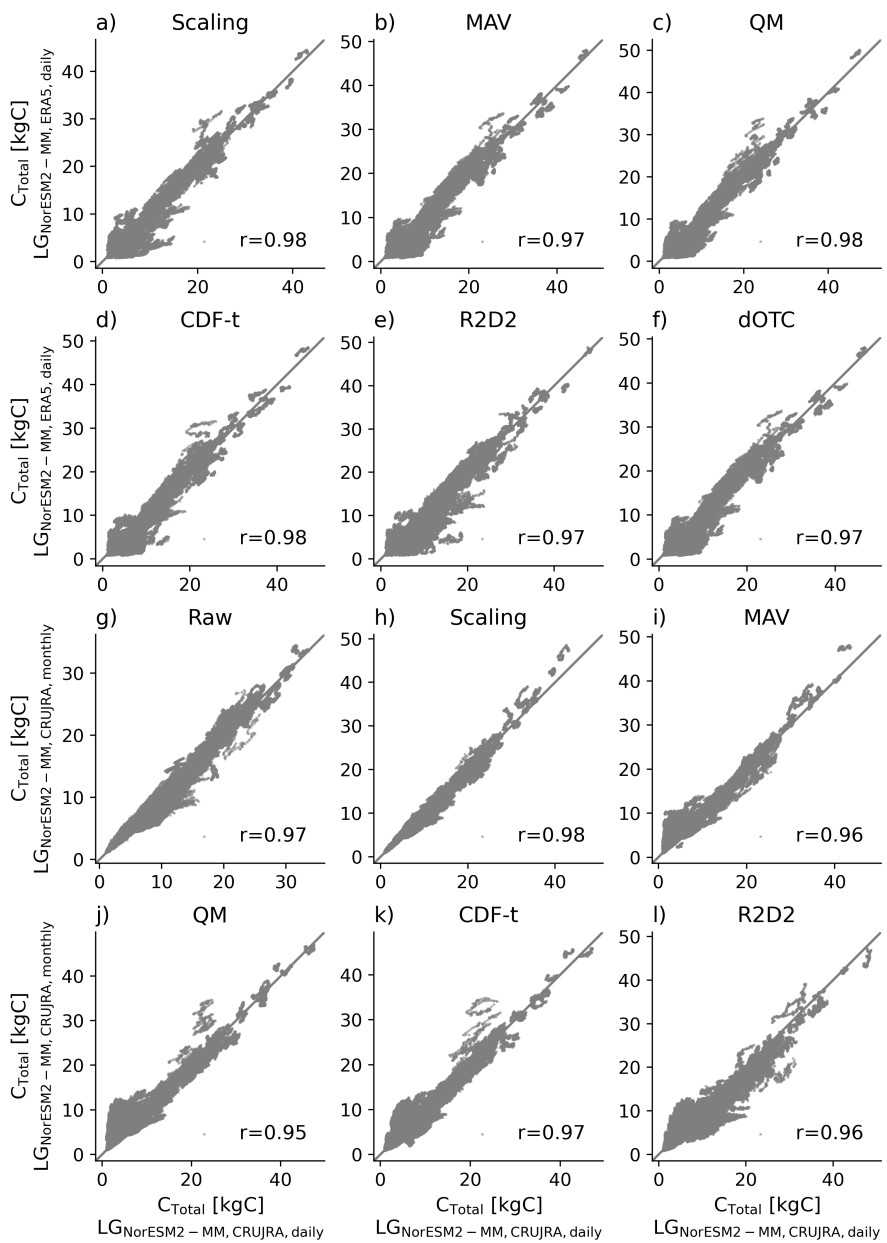

**Figure B7.** Sensitivity to reference dataset (a-e) and to timescale (f-k). Panel a-e show $C_{Total}$ simulated by LPJ-GUESS forced with the NorESM2-MM climate forcing corrected on daily timesteps using the CRUJRA reanalysis as a reference dataset on the x-axis, and $C_{Total}$ simulated by LPJ-GUESS forced with the NorESM2-MM climate forcing corrected on daily timesteps using the ERA5 reanalysis as a reference dataset on the y-axis. Panel f-k show $C_{Total}$ simulated by LPJ-GUESS forced with the NorESM2-MM climate forcing corrected on daily timesteps using the CRUJRA reanalysis as a reference dataset on the x-axis, and $C_{Total}$ simulated by LPJ-GUESS forced with the NorESM2-MM climate forcing corrected on monthly timesteps using the CRUJRA reanalysis as a reference dataset on the y-axis. Each panel also contains the Pearson correlation coefficient ('r).



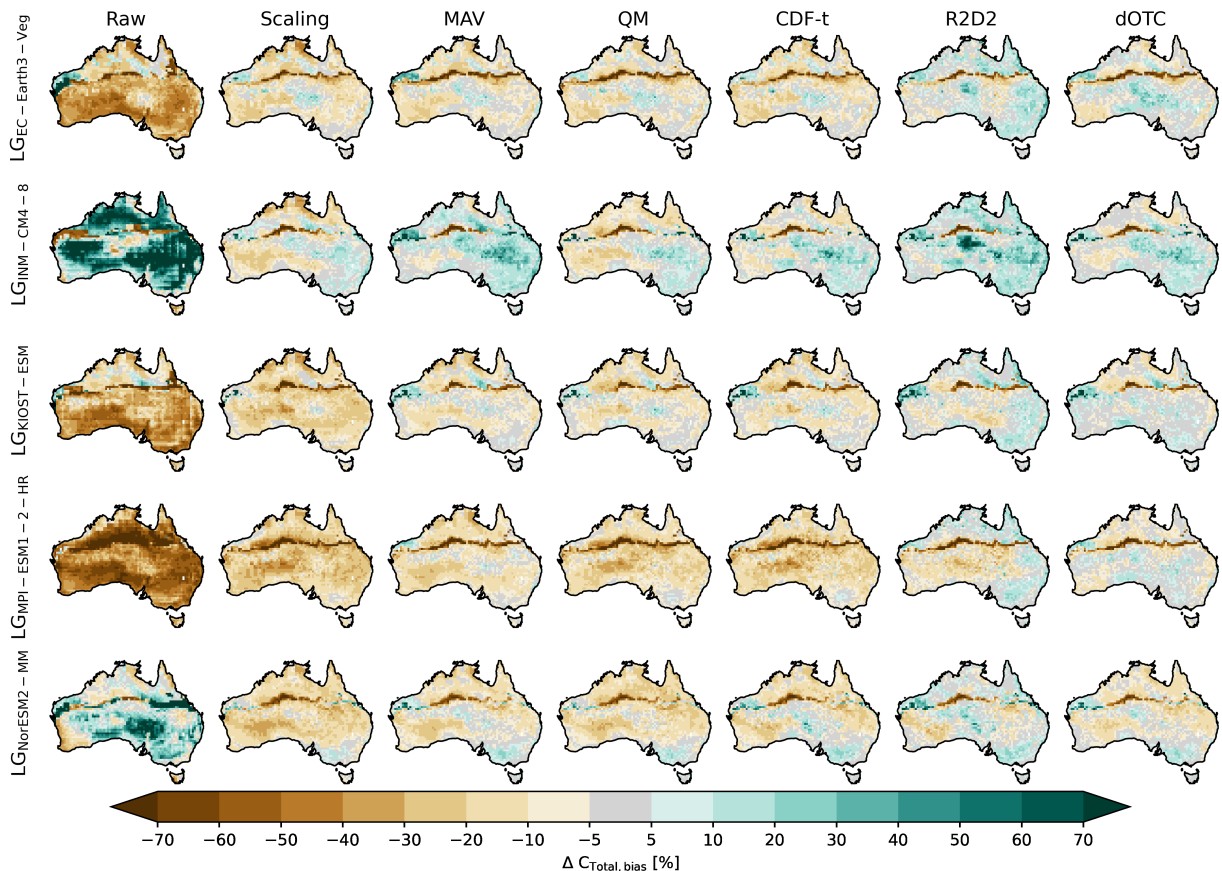

**Figure B8.** Difference between the simulated $C_{Total}$ based on the five bounding models and the CRUJRA reanalysis when LPJ-GUESS is forced with the raw model forcing or with the corrected forcing following the Scaling, MAV, QM, CDF-t, dOTC and R2D2 approach. The bottom row shows the different ensemble averaging methods (arithmetic average, weighted average, and random forest).

*Author contributions.* LT, MGDK, and AJP designed and conducted the experimental analysis. LT, MDK, AJP and GA wrote the first draft with contributions from AMU, SH, BS, and BF. AMU preprocessed the CMIP6 GCM output. BF helped to implement the bias correction methods, GA and SH developed the weighted ensemble averaging method. All authors contributed to the final manuscript. Correspondence and requests for materials should be addressed to LT (l.teckentrup@unsw.edu.au).

*Competing interests.* MDK is a member of the editorial board of Biogeosciences. The authors have no other competing interests to declare.



*Acknowledgements.* The research was funded by the ARC Centre of Excellence for Climate Extremes (CE170100023) and by the New South Wales Department of Planning, Industry and Environment. MDK and AJP acknowledge support from the ARC Discovery Grant (DP190101823). MDK acknowledges support from the NSW Research Attraction and Acceleration Program (RAAP). AMU acknowledges support from the Australian Research Council (DE200100086). We are grateful to the National Computational Infrastructure at the Australian National University for provision of supercomputing resources.




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
