# Peer review of "Opening Pandora's box: Reducing GCM uncertainty in Australian simulations of the carbon cycle"

_EGUsphere, 2022_

## Author Comment (AC2)

Any bias in climate forcing directly influences the model projection of carbon cycle dynamics. Teckentrup et al. quantify the impacts of different bias correction (including univariate correction, multivariate correction, model averages, and random forest method) methods on improving the outputs of carbon stock changes from a dynamic global vegetation model (LPJ-GUESS). This draft was well-written, but I still have some comments on the algorithms used in this analysis, and I think the novelty is insufficient for a paper in ESD.

Major comments:

1. The biggest concern is that after reading I still have no idea which bias correction method should be used to assess the spatial variability or short-term and long-term temporal variation in the total carbon stocks. The results are quite confusing. It would be good to evaluate the classifications of correction methods by function
2. The authors should perform a synthetic analysis and evaluation. The current results are very preliminary.

*We thank the reviewer for their comments. The role of biases in climate forcing is particularly problematic when considering the future of ecosystems that experience the extremes of climate (i.e., very wet, very dry, very hot, very cold). To date, research has focussed on the impact of corrections at global scales, which makes it hard to be sure whether correction approaches affect the nuance of conclusions on regional scales. Our contribution is novel exactly because it tackles a suite of bias correction through the lens of a regional element of the terrestrial carbon cycle that is subject to climate extremes.*

*The reviewer is then making two major comments.*

*First, we interpret the reviewers comment to imply that they were after some sort of ranking of best practice. Our results do not definitively support a single "best" approach. Instead, we show that considering forcing biases is an integral step in carbon cycle projections and has the potential to considerably reduce biases in projections of carbon pools. This is an important step in reducing the large uncertainties in carbon cycle projections. Furthermore, we highlight that bias correction methods have a large potential to reduce carbon cycle uncertainty, namely the simulation of long-term average of carbon pools. Conversely, the bias-correction of forcing data was not able to significantly reduce biases in carbon cycle trends and interannual variability. These findings provide clearer guidance for selecting ensemble weighting and bias correction methods in future studies.*

*Although we did group our evaluation by function, in light of the reviewers comment we will carefully revise to ensure grouping by function is clearer. We will also revise the discussion to better highlight key findings and hope these aspects of the paper will be clearer in the revised version.*

*Second, the reviewer asks for a synthetic analysis. On this point we do not feel this is within the scope of aims of our manuscript. We feel that this type of approach would be more suited for evaluation of individual methods, whereas our paper is trying to synthesise the impact of different approaches through a common lens (LPJ-GUESS and the carbon cycle). We feel that*

*any additional synthetical analysis would simply overcomplicate the story by focussing on very specific details.*

Specific comments:

Fig 1: It would be good to differentiate steps and the name of methods in each step. Can use different icons or colors.

*We will review this figure and the caption to see if we can make it easier to interpret. We will also expand the caption to better explain how steps/methods should be interpreted.*

Table 2: Some of these selected metrics reflect the same (similar) property. For example, all the Root mean squared error, Normalised Mean Error, and Mean bias error indicates the bias in mean value. So the model with good skill in simulating mean value tends to have a higher rank. It is unfair.

*We based the selection of error metrics based on Naughton et al, 2018, who choose these metrics. However, we appreciate the concern of reviewer 1 and will test the ranking using only one of these bias metrics to see how it influences the rankings and revise the manuscript accordingly.*

Ln148: Why use the correlation of 0.3 as a threshold to select the models?

*We agree with the reviewer that the choice of 0.3 as a threshold may seem arbitrary (as would any choice of threshold). Given the selection of independent GCMs is based on the assumption that simulations are independent when their biases are not correlated, we choose 0.3 as a threshold given it is relatively commonly interpreted as the threshold between weak and moderate correlations. We will clarify this in the revised manuscript.*

Ln305-314: Please clarify which meteorological forcing influence the mean value of C_total, the short-term variability (i.e., inter-annual variability) in C_total, and the long-term variability in C-total.

*We thank the reviewer for the suggestion and will adjust the manuscript accordingly. Given Australia is strongly water-limited, both total ecosystem carbon and interannual variability is mostly driven by precipitation (compare Haverd et al., 2013). We also tested this in an additional analysis (which is not shown in the paper) where we only bias-corrected the precipitation data, leaving temperature and radiation as the raw data values. This yielded very similar C_total to runs where all meteorological variables were bias-corrected, suggesting the influence of precipitation dominates in our study region (discussed in "General caveats"). However given bioclimatic limits are prescribed in the model used for this study, temperature will influence the vegetation distribution as well (see below).*

Ln320: In Fig3, only squares and circles indicate a larger bias in mean PPT after multivariate bias correction. Why?

*We agree with the reviewer that this is a surprising result, and note that in fact this behaviour is also apparent for the univariate CDF-t approach. This will require further investigation, and will be discussed in the revisions.*

Ln350: The authors should give a summarized metric showing which bias correction method is better. It is difficult to find the best model by eyes.

*We appreciate the reviewer's suggestion, and will include new text to help guide the reader better by discussing the advantages and disadvantages of each approach.*

The spatial patterns of bias and CV of C_total simulated by the model in Fig 5 and Fig 6 have a clear and strange strip with extreme values. This is not reasonable. Could you please explain why this strip exists?

*In this study, we employed the dynamic global vegetation model LPJ-GUESS. Like many DGVMs (see Fisher et al., 2015), it prescribes bioclimatic limits that define the geographic location growth based on temperature. For example, vegetation growth of $C_4$ grasses and tropical trees is restricted by a lower temperature boundary such that these vegetation types cannot establish or survive when the 20-year average minimum temperature falls below 15.5 degrees Celsius. Therefore, $C_4$ grasses and tropical trees only grow north of the Tropic of Capricorn, while south of it only temperate trees and $C_3$ grasses are simulated. The strong variation across GCMs in simulated temperature thus leads to very different simulated vegetation cover in LPJ-GUESS. Given the boundary between $C_3$ and $C_4$ grasses will depend on the GCM used to force LPJ-GUESS, the type of vegetation in this area varies by the GCM forcing, leading to the large error and uncertainty in simulated carbon. We will discuss this in the revised manuscript.*

Ln374-375: It is not clear why C4 grasses would have a higher CV. The authors did not convince me that this is the real reason.

*We explained the pattern in the comment above, and apologise that this was not made clearer, and will revise the manuscript accordingly.*

Ln379-380: The authors did not explain why the bias in C_total relates to foliar projective cover/ Could you please show the relation between C_total and foliar projective cover? Which factors or processes can influence foliar projective cover in the LPJ-GUESS model?

*We apologise for the lack of clarity in this point and will update the manuscript accordingly. Foliar projective cover can be seen as in indicator for the vegetation growth, which ultimately defines the ecosystem carbon. For example, areas with high foliar projective cover (i.e., trees) will tend to intercept greater PAR and thus assimilate more carbon*

*Simulated foliar projective cover also results from vegetation competition in LPJ-GUESS. This in turn is influenced by the climate and other input datasets: For example, the arid areas of Australia will show strong water limitation (i.e. no precipitation, high temperatures) which create unfavourable conditions for tree growth, so that grasses become more competitive. In the coastal areas, and in the tropics of Australia, water is abundant and allows tree growth at the cost of grass expansion. There are also competitive processes amongst tree species, and C3 and C4 grasses, that are driven by temperature (either dynamically or prescribed), or based*

*on incoming short-wave radiation, i.e., vegetation can be shade-tolerant or shade-intolerant. However, incoming shortwave radiation is not a limiting factor in Australia and can therefore be largely neglected for this study. We will clarify this in the revised manuscript.*

Ln415-417: The peak of seasonal GPP was underestimated a lot. Is this because the peak of meteorological variables (like precipitation or temperature) was underestimated and uncorrected?

*We assume that indeed the lower peak in seasonal GPP is driven by the overall bias in both temperature, and precipitation. However, removing the bias in the input forcing does not achieve a perfect match with the target dataset. This is likely because the bias correction did not consider the spatial domain, i.e., the spatial pattern of vegetation distribution remains similar, only the magnitude in bias is reduced. We will make this clearer in the revised manuscript.*

Ln421-422: The bias in the dry season seems very small. So the effects of correcting data in the dry season may not be very useful?

*We thank the reviewer for the question and will clarify this point in the manuscript. The bias for the individual GCMs may appear relatively low, however the ensemble spread across the full CMIP6 ensemble (shaded grey area) indicates significant uncertainty, and depending on the GCM chosen, dry season GPP can either be zero or reach values over 0.2 PgC which is roughly two third of peak wet season GPP in the target dataset.*

Ln442-445 The introduction of the importance of Australia for the estimation of global land carbon sink should not be in the Discussion. Can put it into the Introduction.

*Yes, we will move the paragraph to the introduction.*

Ln469-470: Don't repeat the results of the analysis in the Discussion.

*We thank the reviewer for the suggestion, and will remove this paragraph.*

References

Fisher, R. A., Muszala, S., Verteinstein, M., Lawrence, P., Xu, C., McDowell, N. G., Knox, R. G., Koven, C., Holm, J., Rogers, B. M., Spessa, A., Lawrence, D., and Bonan, G.: Taking off the training wheels: the properties of a dynamic vegetation model without climate envelopes, CLM4.5(ED), Geosci. Model Dev., 8, 3593–3619, https://doi.org/10.5194/gmd-8-3593-2015, 2015.

Haverd, V., Raupach, M. R., Briggs, P. R., J. G. Canadell., Davis, S. J., Law, R. M., Meyer, C. P., Peters, G. P., Pickett-Heaps, C., and Sherman, B.: The Australian terrestrial carbon budget, Biogeosciences, 10, 851–869, https://doi.org/10.5194/bg-10-851-2013, 2013.

---

## Author Comment (AC3)

Review of egusphere-2022-623

Opening Pandora's box: How to constrain regional projections of the carbon cycle

By Teckentrup et al.

In this study, the authors analyze the impact of varying meteorological forcing obtained from the historical CMIP6 GCMs / ESMs on the historical carbon cycle. More specifically, they asses the impact of the selection of the simulated meteorological forcing on the response of the Australian carbon cycle using different strategies, e.g. bias correction, random-forest approach, ensemble averaging methods, as well as one dynamic global vegetation model, LPJ-GUESS. The authors compare the different methods and report their effect on carbon cycle simulation of LPJ-GUESS in space and time.

The analyzes presented by Treckentrup et al. are very interesting, comprehensive and useful in understanding the impact of different meteorological forcing on the carbon cycle. In some places, the manuscript seems a bit overloaded, making it somewhat more difficult to grasp the full scope of the analyses. Overall, the manuscript is well written. I have a few general comments and a short list of specific comments. Thus, I recommend minor revisions before publication.

*Thanks for the positive review of our manuscript. The "bit overloaded" comment echoes Reviewer 1 and we will address this via improving the clarify and "take-home" messages in our revision.*

General comments:

1. I like how the title reads and the scope of the study, but I find it a bit misleading. As far as I can judge, you are not looking into the projection of the carbon cycle, right? Projection, by definition, means simulating a potential future evolution of the system (e.g. boundary conditions are scenario-driven). Your analysis is based on historical simulations, where we have access to the boundary conditions, e.g. greenhouse gases, volcanic/anthropogenic aerosol loading, etc. So, I would use the word "regional simulations".

   *We thank the reviewer for the suggestions and will adjust the title accordingly.*

   Then I do not really see how you constrain the regional carbon cycle uncertainty. You show that one land model simulates different CC response dependent on different meteorological inputs. In your previous paper, you showed the uncertainty that is related to the variety of land models. So, I would say that you comprehensively demonstrate the entire uncertainty in simulating the carbon cycle related to the choice of models and choice of forcing, but I don't see really how you would go about in constraining this uncertainty. The proposed bias correction methods etc. do not really contribute to reduce the uncertainty, since, if we now ran all TRENDY models with your reanalysis-corrected / or "ensemble average weighting" meteorological forcing, we would end up with a similar uncertainty in the CC response. Bottomline is,

maybe you should focus more on the "full uncertainty" aspect in communicating your analysis, than the "constraining" aspect.

*This is a very insightful comment that we will address in a revised manuscript. The reviewer is of course correct, that to sample the full (or a "fuller") uncertainty would necessitate driving the full TRENDY ensemble with corrected data, perhaps a future research direction – we will make this point clearly in revision. What we are doing is demonstrating how biases in climate forcing can be constrained and what some of the implications may be for the terrestrial carbon cycle.*

2. Overall, I am very surprised that the effect of CO2 on plants, e.g. on water-use efficiency, or the direct stimulation of carbon assimilation, is not being discussed nor mentioned here at all. These effects are vital in simulating the carbon cycle under rising CO2. Were these effects accounted for in the LPJ-GUESS setup? I think so, since almost all runs show an increase in C_total, even those which received a decrease in precipitation and increase in temperature as forcing. How would Australian ecosystems accumulate more carbon under these circumstances? To estimate the impact of meteorological forcing, the CO2 effects might not be essential, but still, these effects need to be addressed and communicated.

*We thank the reviewer for this comment. Indeed, the simulations shown in the manuscript were forced with a transient CO2 (and nitrogen deposition) forcing, and we will add a paragraph in the discussion to clarify this.*

*The climate forcing and other external drivers in LPJ-GUESS will always interact, and therefore the divergent carbon cycle response presented in this manuscript does not isolate the impact of climate forcing on carbon cycle uncertainty alone. However, given all LPJ-GUESS simulations have the same configuration apart from the climate forcing, i.e. the prescribed nitrogen deposition and atmospheric CO2 concentration are identical for all ensemble members, we argue that the experiment set-up is reasonable. Further, any future projections based on offline DGVM runs will be linked to a similar problem linked to inherent complex interactions, and we therefore were aiming to show the impact of climate uncertainty, and correction methods, in a default DGVM configuration. In our forthcoming work we plan to unpack these different drivers more clearly when we consider future simulations, the manuscript is about to be submitted.*

3. I am hesitant to suggest more analysis, since this manuscript already contains a lot of analysis and is a bit over-loaded. So, it is difficult to grasp the entire scope of the manuscript. Are that many supplementary figures needed? I would suggest to assess whether one could reduce some parts in the manuscript, so that it becomes better accessible to the reader and the key messages come across.

*Again, this comment reflects on our need to improve the clarity. We will do this in a revised manuscript, including being stronger in removing supplementary figures.*

4.  But I have to suggest at least one additional analysis point: You only use one realization (r1i1p1f1) of each model. To really get an idea of how the specific GCM compares to reanalysis and other GCMs, one should analyze as many realizations as possible. I would even suggest to get meteorological forcing from grand / large ensembles and one can identify real biases in the model. One realization is not representative for the model, except when some data-assimilation / nudging is conducted (e.g. as in reanalysis). I know it would be too much work for this study, but one should think about it.

*We thank the reviewer for the comment and agree. Indeed, we were initially considering using the CESM large ensemble. However, as the reviewer suggests, the additional DGVM runs would mean significant increase in computation time, which we already aimed to reduce by only applying bias correction methods on a subset of GCMs defining the ensemble spread. It may not be practical to add additional runs and note that the results should be viewed as indicative, rather than aiming to define an exact number that defines carbon cycle uncertainty. Our preference therefore is to address this comment via additional material in the discussion.*

Specific Comments:

L18: What does "and above" mean here? and above global scale?

*We thank the reviewer for pointing out this mistake and will remove 'and above' in the revised manuscript.*

LL85-89: Rather long sentence containing many aspects - can you split it up in at least two separate sentences?

*We thank the reviewer for the suggestion, and we will break up the sentence.*

LL92-94: I don't understand the logic of this sentence. TRENDY models use the identical meteorological forcing and show a large difference in the response of the carbon cycle to the forcing. So, this calls for reducing uncertainty in the land-surface model predictions, rather than the meteorological forcing, no?

*We thank the reviewer for the suggestion and will remove the sentence.*

LL97-98: Can you provide more detail on what first generation and second generation DGVMs refer to?

*First generation DGVMs typically simulate plant communities using a single area-averaged representation of each plant functional type (compare Fisher et al., 2018) while second-generation DGVMs simulate vegetation by individual plants with similar properties, such as age, size, or functional type, together. We thank the reviewer for pointing out that this needs clarification and will update the introduction accordingly.*

L103: What simulation? Please be more specific. It is probably the "historical" simulation, but there are others, like esmHist, where the carbon cycle is fully coupled, etc.

*We thank the reviewer for pointing out the lack of clarity and will include the information about the simulation (the historical simulation of CMIP6 is used here).*

L104: What about the information about atmospheric humidity, i.e. VPD?

*As a legacy of its development and (lack of) availability of humidity data, LPJ-GUESS does not use VPD as an input forcing. Stomatal conductance is based on an empirical boundary layer parameterisation following Huntingford and Monteith, 1998. This parametrisation expresses large-scale evapotranspiration as a hyperbolic dependency on surface resistance (i.e the inverse of stomatal conductance). Therefore, humidity as an input driver is not needed for LPJ-GUESS (compare Smith et al., 2014).*

LL106-7: Can you really do that? Shouldn't you recycle all the inputs consistently then? You can have strong precipitation with simultaneous high shortwave radiation - what does LPJ-GUESS make out of these physically implausible inputs?

*We apologise for the mistake and will rerun the simulations affected following this suggestion. Given incoming shortwave radiation does not limit vegetation growth in Australia, we expect that the results will not change.*

LL108-109: This means you are doing some heavy down-scaling the input variables to a quite high resolution in comparison to the native resolution of the GCMs. Maybe better to remap to a common 1x1 degree grid, no? Or maybe it'd be better to use downscaled CMIP6 output, e.g. https://eartharxiv.org/repository/view/2646/

*We remapped the relatively coarse GCM output to a 0.5 degree given that is the native grid of LPJ-GUESS. While we are aware that dynamically downscaled data exists, such as the CORDEX dataset, or ISIMIP. We chose the CMIP6 forcing given it was the newest climate simulation dataset available and has the largest number of ensemble members. Further, output from the CMIP6 ensemble is commonly used as input drivers for both regional, and global studies of terrestrial ecosystems.*

L125: I think, that is not true. ERA5 is in 0.5x0.5 grid and there is a derivative that is at 0.25x0.25, but 0.05 seems extremely high resolution for reanalysis.

*We apologise for the mistake and will correct that ERA5-Land is on a 0.1 spatial grid.*

Figure 1: I think it would benefit the understanding of Figure 1, if you provided a slightly more elaborate figure caption. At least, you could specify the acronyms used in the figure, so the figure is readable without searching in the text for the acronym definitions.

*We thank the reviewer for the suggestion and will update the figure legend accordingly.*

L140: Can you provide more information on this estimator?

Table 2: The definition of the the summation notation would need more information to be mathematically correct, but I guess it is understandable as it is. https://en.wikipedia.org/wiki/Root-mean-square_deviation

*We thank the reviewer for the comment and will update the equation accordingly.*

L143: Well, these models historically evolved and they share code and concepts. It's hard to define which models are independent. Also, the models that are used to create the reanalysis e.g. IFS for ERA5 share code with CMIP6 models.

*We agree with the reviewer and note that we are aware that this is just one of the many ways to define GCM independence. We will also clarify that dependence to reanalysis datasets can exist.*

L147-148: Also, models that are highly dependent might not "correlate more" on monthly time-scale as the atmosphere is chaotic and highly dependent on the initial state etc.; I would assume that correlation of the spatial pattern in the climatological mean would provide more information. So, I think similar spatial bias matching would give you an idea whether models are similar or not, but maybe you do that, I did not fully understand.

*We thank the reviewer for the suggestion and apologise for the lack of clarity in the methods description. We derive the correlation in the bias over all timesteps, and grid points, and therefore account for spatial patterns in bias.*

L166: "Let us define" ?

*We apologise for the lack of clarity and will update the sentence in the revised manuscript.*

LL170-173: Does this part connect to any paragraph?

*We included a brief description of univariate vs multivariate bias correction methods (205-210) in the methods description to remind the reader what they are.*

L180: If you used temperature in Kelvin scale (so no negative values), one could only use this function for scaling consistently for all variables, no?

*We apologise but do not quite understand this comment and would appreciate if the reviewer clarified it.*

L191: "Let us denote" ?

*We apologise for the lack of clarity and will update the sentence in the revised manuscript.*

L205: Not sure how this fits in the structure of the paragraph.

*We thank the reviewer for the suggestion and will remove both sections on the general difference between univariate and multivariate correction methods.*

LL231-232: Then I really wonder why some representation of atmospheric humidity is not an input to LPJ-GUESS.

*Given LPJ-GUESS focuses primarily on tracking carbon and nitrogen fluxes, the stomatal conductance in simulated vegetation is driven by CO2 and soil moisture rather than VPD. We agree that this is a shortcoming of the model and note that current model development efforts are looking to account for VPD when stomatal conductance is simulated (see Belda et al., 2022).*

LL276-277: Can you explain why you include non-physical parameters such as longitude and latitude in the random-forest approach. Especially for a regional study, I would advise against this practice.

*Testing the performance of the RF model out-of-sample with versus without including geolocation information have shown that including longitude and latitude information improve prediction performance, this is likely because including geolocation predictors enables RF to capture spatial dependencies.*

Figure 2: b,d,f are the same - but I saw the uploaded corrected figure.

*We apologise, and will include the correct figure in the revised manuscript.*

LL305-onwards: Would it make sense to compare carbon fluxes from the actual CMIP6 models to get an estimate for carbon cycle uncertainty? Not all models (e.g. MPI−ESM1−2−HR), but most have some representation of the carbon cycle and the carbon fluxes? I also understand if you only wanted to focus on the effect of the selection of the meteorological forcing.

*We agree that it is important to also consider carbon cycle projections from coupled simulations in CMIP6. However, we here aimed to focus exclusively on uncertainty in the meteorological forcing in offline runs given bias correction methods can be applied which is not possible for coupled runs. We will add this in the discussion.*

Figure 3: "PPT" is a rarely seen abbreviation for precipitation, better pr?

*We thank the reviewer for the suggestion and will update the abbreviation for precipitation.*

LL446-447: In the context of Australia, I would assume one can also add "improved prediction of fire risk", as fire depends largely on the fuel load thus vegetation / carbon cycle.

*We agree with the reviewer and will add the suggestion in the manuscript.*

LL589-590: Counter-argument: One should not only rely on using one DGVM for studies on ecosystem/carbon cycle impact. Maybe you can make the point, that we should use multiple DGVMs and multiple GCMs forcings.

*We agree with the reviewer and will make this point in the discussion.*

References

Huntingford, C., Monteith, J.L. The behaviour of a mixed-layer model of the convective boundary layer coupled to a big leaf model of surface energy partitioning. *Boundary-Layer Meteorology* **88**, 87–101 (1998). https://doi.org/10.1023/A:1001110819090

Smith, B., Wårlind, D., Arneth, A., Hickler, T., Leadley, P., Siltberg, J., and Zaehle, S.: Implications of incorporating N cycling and N limitations on primary production in an individual-based dynamic vegetation model, Biogeosciences, 11, 2027–2054, https://doi.org/10.5194/bg-11-2027-2014, 2014.

---

## Author Response (AR1)

*We thank the reviewer for their insightful comments. In the following, we give our responses (italic black), and highlight modifications made in the revised manuscript (italic red).*

Any bias in climate forcing directly influences the model projection of carbon cycle dynamics. Teckentrup et al. quantify the impacts of different bias correction (including univariate correction, multivariate correction, model averages, and random forest method) methods on improving the outputs of carbon stock changes from a dynamic global vegetation model (LPJ-GUESS). This draft was well-written, but I still have some comments on the algorithms used in this analysis, and I think the novelty is insufficient for a paper in ESD.

Major comments:

1. The biggest concern is that after reading I still have no idea which bias correction method should be used to assess the spatial variability or short-term and long-term temporal variation in the total carbon stocks. The results are quite confusing. It would be good to evaluate the classifications of correction methods by function
2. The authors should perform a synthetic analysis and evaluation. The current results are very preliminary.

*We thank the reviewer for their comments. The role of biases in climate forcing is particularly problematic when considering extremes of climate (i.e., very wet, very dry, very hot, very cold). To date, research has focussed on the impact of corrections at global scales, which makes it hard to assess whether correction approaches affect the nuance of conclusions on regional scales. This is particularly the case in areas subject to unusually variably extreme climate, like Australia. Our contribution is novel exactly because it tackles a suite of bias corrections through the lens of a regional component of the terrestrial carbon cycle that is subject to marked climate extremes.*

*The reviewer is then making two major comments.*

*First, we interpret the reviewers comment to imply that they were after some sort of ranking of best practice. Our results do not definitively support a single "best" approach. Instead, we show that considering forcing biases is an integral step in generating and application of carbon cycle projections and has the potential to considerably reduce biases in projections of carbon pools. This is an important step in reducing the large uncertainties in carbon cycle projections that relate solely to forcing inputs. Furthermore, we highlight that bias correction methods have a large potential to reduce carbon cycle uncertainty, namely the simulation of long-term average of carbon pools. Conversely, the bias-correction of forcing data was not able to significantly reduce biases in carbon cycle trends and interannual variability. These findings provide clearer guidance for selecting ensemble weighting and bias correction methods in future studies.*

*In light of the reviewer's questions about more direction about which approach to take, we have revised our figure groupings to make it clearer how they relate to individual methods. We have also revised our discussion to better highlight key findings and the link to the correction method. Specifically, we now*

1. Explained the advantages of the different bias correction approaches in the discussion (see also below)

   *The methods tested range in complexity. The widely used scaling method applied in this study can correct the mean values of the variables, however, cannot adjust variability and extreme values correctly (see for example Berg et al., 2012). The mean and variance approach therefore builds on the scaling method by correcting both mean and variance. We also considered two alternative approaches that attempt to correct the bias based on their distribution, i.e. quantile mapping and CDF-t. The basic quantile mapping method not only corrects the mean bias but also adjusts the distribution and may therefore be more suitable when both the average and extremes are studied. Based on quantile mapping, the CDF-t method additionally incorporates projected changes in mean and variability simulated by the GCM. In contrast to the univariate approaches discussed, multivariate correction methods allow to adjust intervariable dependencies. One of the main differences between the dOTC and R2D2 methods applied here is that dOTC is designed to transfer some of the multidimensional properties from the GCM to the bias-corrected data (such as the change in time; see François et al., 2021). The R2D2 approach instead assumes that inter-variable and intersite rank correlations are stable in time.*

2. Highlighted the problem of mismatching numbers of days with zero precipitation

   *Lastly, we chose to correct daily climate data for the main analysis. However, correcting monthly data may be statistically more robust, especially for highly variable climate variables with a large number of null values such as daily precipitation. Indeed, our analysis of the corrected input variables surprisingly showed an increase in bias in simulated precipitation for two GCMs after correction (see Fig. 3) which is likely linked to a mismatch in simulated days without rain in the target dataset and the GCM simulation. [...] Given only a few grid cells displayed an unreasonably high bias in precipitation (not shown), and the fact that vegetation growth is also driven by temperature and incoming shortwave radiation in LPJ-GUESS, we assume that the impact on the simulated carbon on monthly-multidecadal timescales is small.*

3. Discussed potential advantages and shortcomings of the different bias correction methods applied

   *In summary, within a framework of testing bias correction methods on the five models spanning the CMIP6 model spread, we found that the bias correction methods successfully reduced the bias to the reference dataset for averages over time and space ($C_{Total}$). Overall, the two multivariate approaches achieved a stronger reduction in bias for both individual GCMs and the ensemble average while also presenting a lower uncertainty across the ensemble. A clear advantage of applying multivariate approaches is that they account for intervariable dependencies and can therefore preserve the consistency between the climate variables used to drive LPJ-GUESS. However, the variation across the different correction methods is small, and value ranges for multivariate are comparable to the univariate quantile mapping*

*approaches. Given the increased computation cost associated with multivariate approaches, and the limited benefit demonstrated in this study, multivariate bias correction methods may therefore not necessarily be the best approach in future impact studies. Further, all correction methods* show limited impact for other temporal properties (such as the change over time; e.g. Hagemann et al., 2011; Maurer and Pierce, 2014; Cannon et al., 2015; François et al., 2020).

*Second, the reviewer asks for a synthetic analysis. We do not think this is within the scope of the aims of our manuscript. This approach would be more suited to evaluate individual methods and their boundary condition behaviour, whereas our paper is trying to synthesise the impact of a range different approaches through a common lens (LPJ-GUESS and the carbon cycle). We feel that synthetic analysis in addition to what's here would unnecessarily overcomplicate the paper's narrative by (a) focussing on very specific, esoteric details whose relevance to the conclusions we present are not clear, and (b) crowd the paper with results (at very least in supplementary material) that are purely theoretical in nature. Moreover, the individual methods have their own assessment papers that perform these kinds of analyses at a level of detail we cannot hope to replicate here, which we cite for the interested reader. We also note that another reviewer suggested the manuscript was already 'overloaded' with analyses. Adding this would clearly not help this perception.*

Specific comments:

Fig 1: It would be good to differentiate steps and the name of methods in each step. Can use different icons or colors.

*We reviewed this figure and the caption to make it easier to interpret. We will also expanded the caption to better explain how steps/methods should be interpreted. We updated the colors and also adjusted the caption to*

*Schematic for study set-up. All terms are defined in the text and the key steps are described in the text. GCM refers to Global circulation models. MAV, QM, CDF-t, dOTC and R2D2 represent five different bias correction methods (Mean and Variance, Quantile Mapping, Cumulative Distribution Function, Dynamical Optimal Transport Correction, and Rank Resampling For Distributions and Dependences, respectively).*

Table 2: Some of these selected metrics reflect the same (similar) property. For example, all the Root mean squared error, Normalised Mean Error, and Mean bias error indicates the bias in mean value. So the model with good skill in simulating mean value tends to have a higher rank. It is unfair.

*We based the selection of error metrics based on Haughton et al. (2018). However, we appreciate the concern of reviewer 1 and excluded the RMSE and NME metrics given they combine the mean, standard deviation, and correlation in the revision. Based on this change, we excluded the GFDL-CM4 GCM and included the MPI-ESM1-2-HR GCM from the subset of skilled GCMs and adjust all figures accordingly. Our results changed only slightly (bias in $C_{Total}$ increased from -18.1 PgC to -18.9 PgC) which did not change our conclusion.*

Ln148: Why use the correlation of 0.3 as a threshold to select the models?

*We agree with the reviewer that the choice of 0.3 as a threshold may seem arbitrary (as would any choice of threshold). Given the selection of independent GCMs is based on the assumption that simulations are independent when their biases are not correlated, we chose 0.3 as a threshold given it is relatively commonly interpreted as the threshold between weak and moderate correlations. We adjusted the text to highlight this shortcoming, and added*

*While 0.3 is an arbitrary threshold, it is commonly interpreted to represent weak to moderate correlation.*

*in the revision.*

Ln305-314: Please clarify which meteorological forcing influence the mean value of C_total, the short-term variability (i.e., inter-annual variability) in C_total, and the long-term variability in C-total.

*We thank the reviewer for the suggestion and will adjust the manuscript accordingly. Given Australia is strongly water-limited, both total ecosystem carbon and interannual variability is mostly driven by precipitation (compare Haverd et al., 2013). We also tested this in an additional analysis (which is not shown in the paper) where we only bias-corrected the precipitation data, leaving temperature and radiation as the raw data values. This yielded very similar $C_{Total}$ to runs where all meteorological variables were bias-corrected, suggesting the influence of precipitation dominates in our study region (discussed in "General caveats").*

Ln320: In Fig3, only squares and circles indicate a larger bias in mean PPT after multivariate bias correction. Why?

*We agree with the reviewer that this is a surprising result and note that in fact this behaviour is also apparent for the univariate CDF-t approach. We investigated this behaviour and found that for the two models, some grid cells show large biases that dominate the increase in bias after correction. In these grid cells, the distribution of daily precipitation is not adjusted perfectly which is likely due to a mismatch in the count of days without precipitation. For example, in one grid cell the target dataset simulates 3167 days without any rainfall while in the corresponding grid cell in the GCM simulation only 62 days are without rain. The adjustment of precipitation, especially on daily timescales, is a known issue in bias correction methods (e.g. Vrac et al., 2016), and future work will need to further research approaches to correct highly variable climate variables such as rainfall. We also noted this shortcoming in the discussion, and added*

*Lastly, we chose to correct daily climate data for the main analysis. However, correcting monthly data may be statistically more robust, especially for highly variable climate variables with a large number of null values such as daily precipitation. Indeed, our analysis of the corrected input variables surprisingly showed an increase in bias in simulated precipitation for two GCMs after correction (see Fig. 3) which is likely linked to a mismatch in simulated days without rain in the target dataset and the GCM simulation. […] Given only a few grid cells displayed an unreasonably high bias in precipitation (not shown), and the fact that vegetation growth is also driven by temperature and incoming shortwave radiation in LPJ-GUESS, we assume that the impact on the simulated carbon on monthly-multidecadal timescales is small.*

Ln350: The authors should give a summarized metric showing which bias correction method is better. It is difficult to find the best model by eyes.

*We appreciate the reviewer's suggestion, and have included new text to help guide the reader better by discussing the advantages and disadvantages of each approach. Given the differences across the different approaches are small, we do not think there is one bias correction method that is superior to the others which is a result in itself. We added in the discussion:*

*The methods tested range in complexity. The widely used scaling method applied in this study can correct the mean values of the variables, however, cannot adjust variability and extreme values correctly (see for example Berg et al., 2012). The mean and variance approach therefore builds on the scaling method by correcting both mean and variance. We also considered two alternative approaches that attempt to correct the bias based on their distribution, i.e. quantile mapping and CDF-t. The basic quantile mapping method not only corrects the mean bias but also adjusts the distribution and may therefore be more suitable when both the average and extremes are studied. Based on quantile mapping, the CDF-t method additionally incorporates projected changes in mean and variability simulated by the GCM. In contrast to the univariate approaches discussed, multivariate correction methods allow to adjust intervariable dependencies. One of the main differences between the dOTC and R2D2 methods applied here is that dOTC is designed to transfer some of the multidimensional properties from the GCM to the bias-corrected data (such as the change in time; see François et al., 2021). The R2D2 approach instead assumes that inter-variable and intersite rank correlations are stable in time.*

The spatial patterns of bias and CV of C_total simulated by the model in Fig 5 and Fig 6 have a clear and strange strip with extreme values. This is not reasonable. Could you please explain why this strip exists?

*In this study, we employed the dynamic global vegetation model LPJ-GUESS. Like many DGVMs (see Fisher et al., 2015), it prescribes bioclimatic limits that define the geographic location growth based on temperature. For example, vegetation growth of $C_4$ grasses and tropical trees is restricted by a lower temperature boundary such that these vegetation types cannot establish or survive when the 20-year average minimum temperature falls below 15.5 degrees Celsius. Therefore, $C_4$ grasses and tropical trees only grow north of the Tropic of Capricorn, while south of it only temperate trees and $C_3$ grasses are simulated. The strong variation across GCMs in simulated temperature thus leads to very different simulated vegetation cover in LPJ-GUESS. Given the boundary between $C_3$ and $C_4$ grasses will depend on the GCM used to force LPJ-GUESS, the type of vegetation in this area varies by the GCM forcing, leading to the large error and uncertainty in simulated carbon.*

*We added in the discussion:*

*We note that in all maps display high values for both bias and CV in $C_{Total}$ across the Tropic of Capricorn (S 23°26'10.7"). This is an artefact resulting from assumed model bioclimatic limits. In LPJ-GUESS, vegetation growth of $C_4$ grasses and tropical trees is restricted by a lower temperature boundary such that these vegetation types cannot establish or survive when the*

*20-year-average minimum temperature falls below 15.5°C. Therefore, $C_4$ grasses and tropical trees only grow north of the Tropic of Capricorn, while south of it only temperate trees and $C_3$ grasses are simulated. The strong variation across GCMs in simulated temperature thus leads to very different simulated vegetation cover (and thus high CV) in LPJ-GUESS.*

Ln374-375: It is not clear why C4 grasses would have a higher CV. The authors did not convince me that this is the real reason.

*We explained the pattern in the comment above, and apologise that this was not made clearer. We revised the manuscript accordingly.*

Ln379-380: The authors did not explain why the bias in C_total relates to foliar projective cover/ Could you please show the relation between C_total and foliar projective cover? Which factors or processes can influence foliar projective cover in the LPJ-GUESS model?

*We apologise for the lack of clarity in this point and updated the manuscript accordingly. Foliar projective cover can be seen as in indicator for the vegetation growth, which ultimately defines the ecosystem carbon. For example, areas with high foliar projective cover (i.e., trees) will tend to intercept greater photosynthetic active radiation and thus assimilate more carbon.*

*Simulated foliar projective cover also results from vegetation competition in LPJ-GUESS. This in turn is influenced by the climate and other input datasets: For example, the arid areas of Australia will show strong water limitation (i.e. no precipitation, high temperatures) which create unfavourable conditions for tree growth, so that grasses (and the $C_4$ grasses in particular due to their higher water-use efficiency) become more competitive. In the coastal areas, and in the tropics of Australia, water is abundant and allows tree growth at the cost of grass expansion. There are also competitive processes amongst tree species, and $C_3$ and $C_4$ grasses, that are driven by temperature (either dynamically or prescribed), or based on incoming short-wave radiation, i.e., vegetation can be shade-tolerant or shade-intolerant. However, incoming shortwave radiation is not a limiting factor in Australia and can therefore be largely neglected for this study. We clarified this in the revised manuscript, and added*

*To examine the impact of bias correction on vegetation composition we examine the FPC which can be seen as indicator for the vegetation growth (due to the relationship between foliar area and light interception), and species competition through tree-grass shading.*

*We further added in the discussion*

*In LPJ-GUESS, FPC results from simulated vegetation competition which in turn is influenced by the climate input forcing. For example, water-limited regions such as arid Australia will have limited tree growth, and increased grass growth. Further, competitive processes amongst tree species, and $C_3$ and $C_4$ grasses, that are driven by temperature (either dynamically or prescribed) can drive vegetation competition and therefore FPC.*

Ln415-417: The peak of seasonal GPP was underestimated a lot. Is this because the peak of meteorological variables (like precipitation or temperature) was underestimated and uncorrected?

*We agree that indeed the lower peak in seasonal GPP appears to be driven by the overall bias in both temperature, and precipitation.*

Ln421-422: The bias in the dry season seems very small. So the effects of correcting data in the dry season may not be very useful?

*We thank the reviewer for the question and clarified this point in the manuscript. The bias for the individual GCMs may appear relatively low, however the ensemble spread across the full CMIP6 ensemble (shaded grey area) indicates significant uncertainty, and depending on the GCM chosen, dry season GPP can either be zero or reach values over 0.2 PgC which is roughly two third of peak wet season GPP in the target dataset.*

*In the revised manuscript, we added*

*Figure 8 shows the seasonal GPP for C4 grasses. All simulations, including $LG_{CRUJRA}$, simulate peak productivity in the wet season and minimum productivity in the dry season (see fig. 8 a) but the uncertainty in simulated seasonal GPP is large (see ensemble spread with values between ~0.1 to 0.4 PgC mon$^{-1}$ at the peak of the wet season, and ~0 to 0.15 PgC mon$^{-1}$ at the peak of the dry season).*

Ln442-445 The introduction of the importance of Australia for the estimation of global land carbon sink should not be in the Discussion. Can put it into the Introduction.

*Yes, we moved the paragraph to the introduction.*

Ln469-470: Don't repeat the results of the analysis in the Discussion.

*We thank the reviewer for the suggestion and removed this paragraph.*

References

Berg, P., Feldmann, H. and Panitz, H.-J. Bias correction of high resolution regional climate model data, Journal of Hydrology, Volumes 448–449, 2012, Pages 80-92, ISSN 0022-1694, https://doi.org/10.1016/j.jhydrol.2012.04.026.

Fisher, R. A., Muszala, S., Verteinstein, M., Lawrence, P., Xu, C., McDowell, N. G., Knox, R. G., Koven, C., Holm, J., Rogers, B. M., Spessa, A., Lawrence, D., and Bonan, G.: Taking off the training wheels: the properties of a dynamic vegetation model without climate envelopes, CLM4.5(ED), Geosci. Model Dev., 8, 3593–3619, https://doi.org/10.5194/gmd-8-3593-2015, 2015.

Haughton, N., Abramowitz, G., Pitman, A. et al. Weighting climate model ensembles for mean and variance estimates. Clim Dyn 45, 3169–3181 (2015). https://doi.org/10.1007/s00382-015-2531-3

Haverd, V., Raupach, M. R., Briggs, P. R., J. G. Canadell., Davis, S. J., Law, R. M., Meyer, C. P., Peters, G. P., Pickett-Heaps, C., and Sherman, B.: The Australian terrestrial carbon budget, Biogeosciences, 10, 851–869, https://doi.org/10.5194/bg-10-851-2013, 2013.

Vrac, M., Noël, T., and Vautard, R. (2016), Bias correction of precipitation through Singularity Stochastic Removal: Because occurrences matter, *J. Geophys. Res. Atmos.*, 121, 5237– 5258, doi:10.1002/2015JD024511.

*We thank the reviewer for their insightful comments. In the following, we give our responses (italic black), and highlight modifications made in the revised manuscript (italic red).*

Review of egusphere-2022-623

Opening Pandora's box: How to constrain regional projections of the carbon cycle

By Teckentrup et al.

In this study, the authors analyze the impact of varying meteorological forcing obtained from the historical CMIP6 GCMs / ESMs on the historical carbon cycle. More specifically, they asses the impact of the selection of the simulated meteorological forcing on the response of the Australian carbon cycle using different strategies, e.g. bias correction, random-forest approach, ensemble averaging methods, as well as one dynamic global vegetation model, LPJ-GUESS. The authors compare the different methods and report their effect on carbon cycle simulation of LPJ-GUESS in space and time.

The analyzes presented by Treckentrup et al. are very interesting, comprehensive and useful in understanding the impact of different meteorological forcing on the carbon cycle. In some places, the manuscript seems a bit overloaded, making it somewhat more difficult to grasp the full scope of the analyses. Overall, the manuscript is well written. I have a few general comments and a short list of specific comments. Thus, I recommend minor revisions before publication.

*Thanks for the positive review of our manuscript. The "bit overloaded" comment echoes Reviewer 1 and we addressed this via refining out text and improving the clarify and "take-home" messages in our revision.*

General comments:

1. I like how the title reads and the scope of the study, but I find it a bit misleading. As far as I can judge, you are not looking into the projection of the carbon cycle, right? Projection, by definition, means simulating a potential future evolution of the system (e.g. boundary conditions are scenario-driven). Your analysis is based on historical simulations, where we have access to the boundary conditions, e.g. greenhouse gases, volcanic/anthropogenic aerosol loading, etc. So, I would use the word "regional simulations".

   *We thank the reviewer for the suggestions and have adjusted the title accordingly.*

   Then I do not really see how you constrain the regional carbon cycle uncertainty. You show that one land model simulates different CC response dependent on different meteorological inputs. In your previous paper, you showed the uncertainty that is related to the variety of land models. So, I would say that you comprehensively demonstrate the entire uncertainty in simulating the carbon cycle related to the choice of models and choice of forcing, but I don't see really how you would go about in

constraining this uncertainty. The proposed bias correction methods etc. do not really contribute to reduce the uncertainty, since, if we now ran all TRENDY models with your reanalysis-corrected / or "ensemble average weighting" meteorological forcing, we would end up with a similar uncertainty in the CC response. Bottomline is, maybe you should focus more on the "full uncertainty" aspect in communicating your analysis, than the "constraining" aspect.

*The reviewer is correct, to sample the full (or a "fuller") uncertainty would necessitate driving the full TRENDY ensemble with corrected data, perhaps a future research direction – we made this point clearly in revision. What we are doing is demonstrating how biases in one important pathway of the chain - climate forcing - can be constrained and what some of the implications may be for the terrestrial carbon cycle. By using a single model we can ensure a clear attribution pathway. In forthcoming work, we also focus on resolving parameter uncertainty, alongside these constrained climate forcing, to consider the broader implications.*

*In revision we made this distinction clearer, and added in the discussion*

*In addition, Teckentrup et al. (2021) showed significant uncertainty in the simulated terrestrial carbon cycle linked to the choice of DGVM, but in this study we chose a single DGVM to study the impact of climate uncertainty. However, to capture the full uncertainty, and to achieve a stronger constraint on the simulated terrestrial carbon cycle, future work could explore the response in other members of the TRENDY ensemble, and create an ensemble composed of both different DGVMs and different GCM climate forcings.*

2. Overall, I am very surprised that the effect of CO2 on plants, e.g. on water-use efficiency, or the direct stimulation of carbon assimilation, is not being discussed nor mentioned here at all. These effects are vital in simulating the carbon cycle under rising CO2. Were these effects accounted for in the LPJ-GUESS setup? I think so, since almost all runs show an increase in C_total, even those which received a decrease in precipitation and increase in temperature as forcing. How would Australian ecosystems accumulate more carbon under these circumstances? To estimate the impact of meteorological forcing, the CO2 effects might not be essential, but still, these effects need to be addressed and communicated.

*We thank the reviewer for this comment. Indeed, the simulations shown in the manuscript were forced with a transient $CO_2$ (and nitrogen deposition) forcing, and we added a paragraph in the discussion to clarify this.*

*The climate forcing and the rising $CO_2$ in LPJ-GUESS will always interact, and therefore the divergent carbon cycle response presented in this manuscript are not strictly due to the climate forcing alone. However, given all LPJ-GUESS simulations have the same configuration (i.e. the prescribed nitrogen deposition and atmospheric $CO_2$ concentration are identical for all ensemble members) apart from the climate forcing, we argue that the experiment set-up is reasonable. In our forthcoming work we plan to unpack these different drivers more clearly when we consider future simulations, the manuscript is*

*about to be submitted.*

*In the revised manuscript we added a section about the CO₂ and nitrogen deposition forcing in the methods*

*Atmospheric CO$_2$ forcing and nitrogen deposition*

*In addition to the climate forcing, both atmospheric CO$_2$ concentration and nitrogen deposition are transient. We force LPJ-GUESS with the atmospheric CO$_2$ forcing following historical data until the year 2014. For the remaining years, values for the shared socio-economic pathway SSP245 are used (both from Meinshausen et al., 2020). We further prescribe historical nitrogen deposition until 2009. After 2009, LPJ-GUESS is forced with the nitrogen deposition following the representative concentration pathway RCP4.5 (based on Lamarque et al., 2013).*

*We further added in the discussion*

*In addition, the response of the simulated terrestrial carbon cycle to the climate forcing is intimately linked to the sensitivity to the atmospheric CO$_2$ concentration. This study chose a model set-up with both transient atmospheric CO$_2$ concentration and nitrogen deposition, and therefore does not fully isolate the impact of the climate forcing. However, given all LPJ-GUESS simulations have the same configuration apart from the climate forcing, i.e. the prescribed atmospheric CO$_2$ concentration and nitrogen deposition are identical, we argue that our experiment set-up is suitable for this study.*

3. I am hesitant to suggest more analysis, since this manuscript already contains a lot of analysis and is a bit over-loaded. So, it is difficult to grasp the entire scope of the manuscript. Are that many supplementary figures needed? I would suggest to assess whether one could reduce some parts in the manuscript, so that it becomes better accessible to the reader and the key messages come across.

   *Again, this comment reflects on our need to improve the clarity. We accordingly removed the supplementary figures showing the sensitivity to timescale and choice of target dataset.*

4. But I have to suggest at least one additional analysis point: You only use one realization (r1i1p1f1) of each model. To really get an idea of how the specific GCM compares to reanalysis and other GCMs, one should analyze as many realizations as possible. I would even suggest to get meteorological forcing from grand / large ensembles and one can identify real biases in the model. One realization is not representative for the model, except when some data-assimilation / nudging is conducted (e.g. as in reanalysis). I know it would be too much work for this study, but one should think about it.

   *We thank the reviewer for the comment and agree. Indeed, we were initially considering using the CESM large ensemble. However, as the reviewer suggests, the additional DGVM runs would mean significant increase in computation time, which we already aimed to reduce by only applying bias correction methods on a subset of GCMs defining the ensemble spread. We don't feel it is practical to add additional runs and note that the results should be viewed as indicative, rather than aiming to define an exact number that*

*defines carbon cycle uncertainty. Nevertheless, we agree this is a valid point and we added this to our discussion.*

*Further, in this study, we chose just one realization from each GCM, and therefore the results presented in this study do not fully reflect the uncertainty in simulations of the terrestrial carbon cycle linked to the entire spectrum of possible GCM forcings. Adding more realizations would significantly increase the computational costs, and we do not expect that our results would differ significantly. Ukkola et al. (2020) looked at the effects of additional ensemble members in their assessment of future rainfall change and found limited sensitivity. Nevertheless, to fully understand the impact of uncertainty in simulated climate within individual GCMs, future work could consider using the CESM large ensemble.*

Specific Comments:

L18: What does "and above" mean here? and above global scale?

*We thank the reviewer for pointing out this mistake and removed 'and above' in the revised manuscript.*

LL85-89: Rather long sentence containing many aspects - can you split it up in at least two separate sentences?

*We thank the reviewer for the suggestion, and we broke up the sentence.*

LL92-94: I don't understand the logic of this sentence. TRENDY models use the identical meteorological forcing and show a large difference in the response of the carbon cycle to the forcing. So, this calls for reducing uncertainty in the land-surface model predictions, rather than the meteorological forcing, no?

*We thank the reviewer for the suggestion and removed the sentence.*

LL97-98: Can you provide more detail on what first generation and second generation DGVMs refer to?

*First generation DGVMs typically simulate plant communities using a single area-averaged representation of each plant functional type (compare Fisher et al., 2018) while second-generation DGVMs simulate vegetation by individual plants with similar properties, such as age, size, or functional type, together. We thank the reviewer for pointing out that this needs clarification and updated the introduction accordingly.*

*LPJ-GUESS is the only second-generation DGVM part of the TRENDY ensemble, i. e. it is a cohort-based DGVM that incorporates simplified dynamics of forest-gap models. It can therefore be expected to simulate more realistic temporal carbon dynamics than first-generation DGVMs which typically rely on a single area-averaged representation of each plant functional type (PFT) for each climatic grid cell (e.g. Fisher et al., 2018).*

L103: What simulation? Please be more specific. It is probably the "historical" simulation, but

there are others, like esmHist, where the carbon cycle is fully coupled, etc.

*We thank the reviewer for pointing out the lack of clarity and included the information about the simulation (the historical simulation of CMIP6 is used here).*

L104: What about the information about atmospheric humidity, i.e. VPD?

*As a legacy of its development and (lack of) availability of humidity data, LPJ-GUESS does not use VPD as an input forcing. Stomatal conductance is based on an empirical boundary layer parameterisation following Huntingford and Monteith, 1998. This parametrisation expresses large-scale evapotranspiration as a hyperbolic dependency on surface resistance (i.e the inverse of stomatal conductance). Therefore, humidity as an input driver is not needed for LPJ-GUESS (compare Smith et al., 2014).*

LL106-7: Can you really do that? Shouldn't you recycle all the inputs consistently then? You can have strong precipitation with simultaneous high shortwave radiation - what does LPJ-GUESS make out of these physically implausible inputs?

*We apologise for the mistake and reran the simulations affected following this suggestion. The updated runs did not change the results.*

LL108-109: This means you are doing some heavy down-scaling the input variables to a quite high resolution in comparison to the native resolution of the GCMs. Maybe better to remap to a common 1x1 degree grid, no? Or maybe it'd be better to use downscaled CMIP6 output, e.g. https://eartharxiv.org/repository/view/2646/

*We remapped the relatively coarse GCM output to a 0.5 degree given that is the native grid of LPJ-GUESS. While we are aware that dynamically downscaled data exists, such as the CORDEX dataset, or ISIMIP, we chose the CMIP6 forcing given it was the newest climate simulation dataset available and has the largest number of ensemble members. Further, output from the CMIP6 ensemble is commonly used as input drivers for both regional, and global studies of terrestrial ecosystems.*

L125: I think, that is not true. ERA5 is in 0.5x0.5 grid and there is a derivative that is at 0.25x0.25, but 0.05 seems extremely high resolution for reanalysis.

*We apologise for the mistake and will correct that ERA5-Land is on a 0.1 spatial grid.*

Figure 1: I think it would benefit the understanding of Figure 1, if you provided a slightly more elaborate figure caption. At least, you could specify the acronyms used in the figure, so the figure is readable without searching in the text for the acronym definitions.

*We thank the reviewer for the suggestion and updated the figure legend accordingly.*

*Schematic for study set-up. All terms are defined in the text and the key steps are described in the text. GCM refers to Global circulation models. MAV, QM, CDF-t, dOTC and R2D2 represent five different bias correction methods (Mean and Variance, Quantile Mapping, Cumulative Distribution Function, Dynamical Optimal Transport Correction, and Rank Resampling For*

*Distributions and Dependences, respectively).*

L140: Can you provide more information on this estimator?

We used this estimator to derive the optimal number of bins since it is robust to outliers. In the revision, we included a reference for this estimator.

Table 2: The definition of the the summation notation would need more information to be mathematically correct, but I guess it is understandable as it is. https://en.wikipedia.org/wiki/Root-mean-square_deviation

*We thank the reviewer for the comment. Based on a comment from reviewer one, we removed the RMSE as a metric but updated the MBE following this suggestion.*

L143: Well, these models historically evolved and they share code and concepts. It's hard to define which models are independent. Also, the models that are used to create the reanalysis e.g. IFS for ERA5 share code with CMIP6 models.

*We agree with the reviewer and note that we are aware that this is just one of the many ways to define GCM independence. We will also clarify that dependence to reanalysis datasets can exist, and added in the methods*

*We further note that multiple approaches exist to define GCM dependence (see for example Knutti et al., 2017) and following a different method may yield a different result. Moreover, reanalysis products and GCMs can share modules as well which further complicates achieving an estimate of truly independent GCMs.*

L147-148: Also, models that are highly dependent might not "correlate more" on monthly time-scale as the atmosphere is chaotic and highly dependent on the initial state etc.; I would assume that correlation of the spatial pattern in the climatological mean would provide more information. So, I think similar spatial bias matching would give you an idea whether models are similar or not, but maybe you do that, I did not fully understand.

*We thank the reviewer for the suggestion and apologise for the lack of clarity in the methods description. We derived the correlation in the bias over all timesteps, and grid points, and therefore account for spatial patterns in bias.*

L166: "Let us define" ?

*We apologise for the lack of clarity and updated the sentence in the revised manuscript.*

LL170-173: Does this part connect to any paragraph?

*We included a brief description of univariate vs multivariate bias correction methods (205-210) in the methods description to remind the reader what they are. Following this question, we removed this part.*

L180: If you used temperature in Kelvin scale (so no negative values), one could only use this function for scaling consistently for all variables, no?

*We apologise but do not quite understand this comment and would appreciate if the reviewer clarified it.*

L191: "Let us denote" ?

*We apologise for the lack of clarity and will update the sentence in the revised manuscript.*

L205: Not sure how this fits in the structure of the paragraph.

*We thank the reviewer for the suggestion and removed both sections on the general difference between univariate and multivariate correction methods.*

LL231-232: Then I really wonder why some representation of atmospheric humidity is not an input to LPJ-GUESS.

*LPJ-GUESS was originally developed when forcing for VPD was less commonly available and this is a legacy of the model. A new release now has a direct dependency on VPD (see Belda et al., 2022) but we note there is limited differences in simulated carbon fluxes.*

LL276-277: Can you explain why you include non-physical parameters such as longitude and latitude in the random-forest approach. Especially for a regional study, I would advise against this practice.

*Testing the performance of the RF model out-of-sample with versus without including geolocation information have shown that including longitude and latitude information improve prediction performance, this is likely because including geolocation predictors enables RF to capture spatial dependencies.*

Figure 2: b,d,f are the same - but I saw the uploaded corrected figure.

*We apologise, and included the correct figure in the revised manuscript.*

LL305-onwards: Would it make sense to compare carbon fluxes from the actual CMIP6 models to get an estimate for carbon cycle uncertainty? Not all models (e.g. MPI−ESM1−2−HR), but most have some representation of the carbon cycle and the carbon fluxes? I also understand if you only wanted to focus on the effect of the selection of the meteorological forcing.

*We agree that it is important to also consider carbon cycle projections from coupled simulations in CMIP6. However, we here aimed to focus exclusively on uncertainty in the meteorological forcing in offline runs given bias correction methods can be applied which is not possible for coupled runs.*

Figure 3: "PPT" is a rarely seen abbreviation for precipitation, better pr?

*We thank the reviewer for the suggestion and updated the abbreviation for precipitation.*

LL446-447: In the context of Australia, I would assume one can also add "improved prediction of fire risk", as fire depends largely on the fuel load thus vegetation / carbon cycle.

*We agree with the reviewer and added the suggestion in the manuscript.*

LL589-590: Counter-argument: One should not only rely on using one DGVM for studies on ecosystem/carbon cycle impact. Maybe you can make the point, that we should use multiple DGVMs and multiple GCMs forcings.

*We agree with the reviewer and made this point in the discussion.*

*In addition, Teckentrup et al. (2021) showed significant uncertainty in the simulated terrestrial carbon cycle linked to the choice of DGVM, but in this study we chose a single DGVM to study the impact of climate uncertainty. However, to capture the full uncertainty, and to achieve a stronger constraint on the simulated terrestrial carbon cycle, future work could explore the response in other members of the TRENDY ensemble, and create an ensemble composed of both different DGVMs and different GCM climate forcings as well as coupled carbon cycle simulations.*

References

Huntingford, C., Monteith, J.L. The behaviour of a mixed-layer model of the convective boundary layer coupled to a big leaf model of surface energy partitioning. *Boundary-Layer Meteorology* **88**, 87–101 (1998). https://doi.org/10.1023/A:1001110819090

Smith, B., Wårlind, D., Arneth, A., Hickler, T., Leadley, P., Siltberg, J., and Zaehle, S.: Implications of incorporating N cycling and N limitations on primary production in an individual-based dynamic vegetation model, Biogeosciences, 11, 2027–2054, https://doi.org/10.5194/bg-11-2027-2014, 2014.

---

## Author Response (AR2)

**Reviewer 1**

Teckentrup et al. made a lot of efforts to improve this research. Super nice! But I still have some minor comments at the current stage.

Abstract

Ln 8-9: The sentence "Carbon pools are insensitive to the type of bias correction method" is unclear to me. Do you mean "model carbon outputs before and after bias correction are similar" or "bias correction can improve model carbon output and different methods show similar outputs"?

We thank the reviewer for the comment and apologise for the confusion. The latter is correct; we have updated the abstract accordingly.

Ln11-12: "Some bias correction methods reduce the ensemble uncertainty more than others." I cannot extract any useful information from this sentence. Please describe some more clear findings. Which method can reduce more uncertainty? How much uncertainty can be reduced?

We apologise for the lack of clarity and now include more details in the abstract.

> *Multivariate bias correction methods tend to reduce the uncertainty more than univariate approaches, although the overall magnitude was similar.*

Ln16: I would suggest to re-write the result part of the abstract.

We thank the reviewer for the suggestion and revised the abstract as noted below.

> *Even after correcting the bias in the meteorological forcing dataset, the simulated vegetation distribution shows different patterns when different GCMs are used to drive LPJ-GUESS. [...] This highlights that where possible, an arithmetic ensemble average should be avoided. However, potential target datasets that would facilitate the application of machine learning approaches, i.e., that cover both the spatial and temporal domain required to derive a robust informed ensemble average are sparse for ecosystem variables.*

Main text

Ln 74: Weighting methods should be the third strategy to reduce uncertainty, right? The hypothesis of this method is that the uncertainty of different models can cancel each other out

Weighting approaches are based on similar principles to GCM subselection approaches, i.e., the goal is to derive 'representative' ensemble statistics given model ensembles such as CMIP are ensembles of opportunity. Weighting methods **can** be the third strategy and have been shown to outperform simple ensemble averages (even after a subselection is chosen). However, they depend on the availability of suitable target datasets to derive weights from.

Ln 187: Confusing. Here, the authors said, "The bias correction was then applied to each calendar month". But in Ln 180, they said, "We show the correction based on daily timescales"

We apologise for the confusion and will clarify in the revised manuscript. For all bias correction approaches shown in the main manuscript, a correction term is derived and applied on a daily timestep for each month separately. We also tested the impact of the temporal resolution during the bias correction where in a sensitivity experiment, we derived and applied the bias correction on a monthly timestep covering all timesteps (instead of splitting the timeseries up into smaller slices). We revised the methods section accordingly

> *The projection period was split into ten 25-year slices. The bias correction was then derived and applied to each calendar month* *on a daily timestep* *within each time slice separately.*

Ln 276: "The error correlation coefficient is used as …" or "We use the error correlation coefficient…", "We derive the linear combination…" rather than "This method derives…"

We thank the reviewer for the suggestion and updated the revised manuscript accordingly.

Ln300: The sentence "While not all possible combinations of approaches were examined, we employed a wide range of methods" can be removed.

We thank the reviewer for the suggestion and updated the revised manuscript accordingly.

Figure 3: In panel (a). the biases of raw individual model data show most models having positive bias (except for MPI-ESM1-2-HR), so the arithmetic average value should be positive. But why the skill arithmetic average of precipitation bias in panel (a) is negative? Does this mean that the selected models are not representative?

Ln348: The arithmetic ensemble average of the biases in C_total, right?

In line 348 we refer to all variables presented in figure 3, i. e. average precipitation, temperature, and $C_{Total}$.

Ln391: why are no random forest and weighted ensemble results in Figure 6? These two ensembles with the lowest climate bias should have a good performance in predicting C_total.

We did not include the coefficient of variation for the independence or random forest weighted estimates because they produce a single estimate rather than an ensemble estimate. As such, there is no coefficient of variation across the ensemble, just a 'best estimate'. In the revised paper, we add this into the figure caption:

> *Note that we do not show a coefficient of variation for the weighted ensemble averages. Given they produce a single estimate rather than an ensemble estimate, a coefficient of variation does not exist for these methods.*

Figure 8: GPP seasonality forced by the R2D2 method corrected climate data always have the lowest bias. This would be useful information for readers

We thank the reviewer for the suggestion, and included in the revised manuscript

> *Notably, the R2D2 method always achieves the lowest bias to the target dataset compared to the remaining bias correction methods.*

**Reviewer 2**

Thank you for inviting me to review paper: "Opening Pandora's box: How to constrain regional simulations of the carbon cycle" by Teckentrup et al.

First, can I apologise for taking three weeks to return this review. It is me who has held things up – the start to 2023 has involved more meetings than is ideal.

I really like the messages of manuscript, and they are important – anything that provides concise information on ESM biases is important. And to my knowledge, there are few studies that frame near-surface meteorology uncertainty in the context of its impact on the terrestrial carbon cycle. However, I am sorry to say that I think the current write-up of the results needs substantial attention. There is little correlation in places between what the paper achieves, and how its findings are described in the text.

First, what does the paper actually do? If I have understood correctly, then a single land surface model (LPJ-GUESS) is forced with many models from the CMIP6 climate model ensemble. Then various methods are used to remove biases in the CMIP6 models' meteorological output, by relaxing back (via different ways) to the CRUJRA climate data. The resultant scaled climatologies are then used to force, again, the LPG-GUESS DGVM. Changes in predicted features of the terrestrial carbon cycle are analyzed, with an emphasis on Australia.

In light of the above, my starting point was the title and Abstract, but a vagueness meant a continuous jumping forward and backward within the paper to make sure my interpretation was valid. Hence, the following changes would help at the paper start:

(1) Title. Avoid an odd title ("Pandora's box"), and state clearly what the paper is about. First, the paper does not cover multiple "regional" locations – it is almost exclusively about Australia, and second and more importantly, I initially thought the paper was about constraining models of the terrestrial carbon cycle. (I was anticipating some sort of analysis of the TRENDY ensemble of DGVMs, given mention of "the carbon cycle"). In fact, the paper is almost exclusively about bias-correcting ESM outputs and assessing its impact on a single uncorrected land model.

We thank the reviewer for the feedback. We here chose Australia as a test bed for other water-limited regions but appreciate the title may be misleading and accordingly changed it to reflect the focus on Australia. We have opted to retain other elements of the title. Given the strong focus on the simulated carbon cycle, and the fact that referee 1 explicitly stated they appreciated the title, we updated the title to:

*Opening Pandora's box: Reducing GCM uncertainty in Australian simulations of the carbon cycle*

While we agree that assessing the uncertainty linked to different terrestrial biosphere modelling framework is crucial, the impact of the climate forcing on the emerging carbon cycle response has been proven to contribute strongly to carbon cycle uncertainty in previous research (i.e. Ahlström et al., 2012 and 2017; Wu et al., 2017). Indeed, Fig. 2 in the manuscript presents an uncertainty in simulated $C_{Total}$ of 30-70 PgC which is comparable to the uncertainty in $C_{Veg}+C_{Soil}$ across the TRENDY ensemble of roughly 12-90 PgC (see Teckentrup et al., 2021). This emphasizes the need to constrain the impact of climate uncertainty on the simulated carbon cycle. Choosing a single model, in this case LPJ-GUESS, as a representative DGVM of the TRENDY ensemble, allows this manuscript to specifically focus on the impact of climate biases on the carbon cycle/ vegetation response. It is not possible to achieve this using the TRENDY ensemble which was forced using a single meteorological dataset. In the updated manuscript, we updated the introduction to make this point clear:

> *Using a single model forced with multiple realisations of climate us to separate climate-driven uncertainties from those arising from model.*

(2) In the Abstract, as LPG-GUESS itself is not corrected in any way, then this does not provide an overall constraint on land carbon cycle projections. That's fine, as ESM correcting is important, but this needs to be made clear. The confusion occurs with wordings such as: "None of the bias correction methods consistently improve the change in carbon over time". At first reading, I was expecting a comparison again carbon pool datasets – possible some sort of EO product. But the "data" is really LPG-GUESS projections?

Yes, we did not correct LPJ-GUESS. A detailed parametrisation and correction of process representations in any DGVM are not trivial and this was not the focus of our manuscript. This is explicitly mentioned in the methods ('We adopted the global configuration of the model'). Instead, our aim was to get a sense of the impact of uncertainty in the climate datasets used to drive any DGVM in Australia, or, more generally, water limited regions. Therefore, a tuned DGVM is not necessary to understand the effect of climate uncertainty on carbon cycle simulations.

We also note that we are aware that the target datasets chosen in this study, i.e. LPJ-GUESS simulations driven by reanalysis, are not equivalent to observation datasets of any kind. Since we wanted to analyse the impact of the methods chosen to deal with biases rather than achieving a constrained estimate of the simulated carbon cycle (which indeed would need a 'corrected' version of LPJ-GUESS), a somewhat synthetic experiment set up is suitable for this manuscript.

However, we appreciate the reviewer's concern and we have updated ambiguous sections in the revised manuscript. We also revised the abstract

*These biases have been identified as a major source of uncertainty in carbon cycle projections, hampering predictive capacity. In this study we examine different methods to reduce uncertainty in simulations of the carbon cycle in Australia arising from biases in climate projections.*

(3) At multiple locations, the beginning of the paper talks about geographical variation. For instance, "especially at regional scales, climate projections display large biases…". Again, I think the authors need to be clear that the focus of this paper is almost exclusively on Australia.

We thank the reviewer for their comment and note that we chose Australia as a test bed for water-limited regions. To address this concern, we updated the revised manuscript accordingly and clarify we are focussing on Australia as opposed to several different regions globally. We also now mention Australia in the title as noted previously. We still use the term 'regional' in the paper as Australia is a huge continent with many different climate zones, vegetation types etc. and there is a clear need to better understand carbon cycle uncertainties at regional scales within Australia.

The above points only refer to the start of the paper. But this consistent need for clarity as to what the paper achieves need to follow through the entire manuscript.

We have made changes throughout the paper to clarify the text as the reviewer can hopefully see from the tracked changes document.

The authors raise an important point in the Abstract that it may be necessary to "account for temporal properties in correction or ensemble averaging methods". Unfortunately, a lot of the time-evolving issues are lost in the presentation. Figure 2 makes it very clear that removal of an overall invariant bias (approximated by setting all changes to be such that they are zero in year 1920) fails to remove the subsequent large range of gradients in the years out to 2010. Yet after bias correction, we cannot see the individual time-evolving paths – because in Figure 4, if I understand things OK, the individual lines are more a form of ensemble mean. And this will, by definition, remove the spread of gradients.

We thank the reviewer for the feedback and apologise for the confusion. Figure 4 shows the change in CTotal over time for both ensemble averaging methods (Fig. 4f) and for individual simulations based on the five bounding GCMs (Fig. 4a-e). In the revised manuscript, we updated the figure caption

*Figure 4. 30-year moving average of the change in CTotal. In each panel, the bold black line is the change in CTotal obtained using the CRUJRA reanalysis and the grey shaded area represents the full unconstrained CMIP6 model ensemble. Panel a–e show the CTotal change simulated using input from the five individual bounding models separately. The colors show the change in CTotal based on the different bias correction methods. Panel f shows the change in CTotal estimated by the ensemble averaging methods.*

The issue with eventual removing trends (so not just offsets) is that this will in effect give each climate model the same transient climate sensitivity, or even the same equilibrium climate sensitivity; ECS. Hence this will make each ESM warm at roughly similarly rates to each other, for the same GHG pathway, as based on known historical warming. Of course, determining the

true ECS is the planet is the number one task of climate research. However, the reason this is not a trivial task, i.e. simply fitting to historical temperature trends, is because we do not know if we are living in a world with a high ECS value, but aerosols are currently offsetting much warming – or the opposite. The authors should be aware of these issues, because simply adding a correction to gradients based on the historical period could still cause major errors when estimating out to year 2100. This might be worth stating in the Discussion part.

We thank the reviewer for their comment. Firstly, to be clear, while we agree with this comment, it has no qualitative implications for what we present here. While this issue could affect some bias correction approaches, those presented here to do not explicitly correct trends. We have nevertheless noted this issue in the discussion:

> Conversely, explicitly bias correcting trends based on historical data, when the spatiotemporal nature may not yet have clearly emerged, could equally be problematic for unbiased estimation of climate system properties like equilibrium climate sensitivity.

I would link the Abstract two sentences "Some bias correction methods…." and "The vegetation distribution…." – make the same sentence? Because this is a key point of the manuscript, that when scanning across a range of bias-correction possibilities, major DGVM projection differences remain. Such differences are sufficiently large that even for a single land surface response, that DGVM can estimate alternative vegetation distributions. (This statement then encourages the reader to consider in detail Figure 7).

Following this Reviewer's suggestion and feedback from referee 1, the abstract now reads

> Multivariate bias correction methods tend to reduce the uncertainty more than univariate approaches, although the overall magnitude is similar. Even after correcting the bias in the meteorological forcing dataset, the simulated vegetation distribution presents different patterns when different GCMs are used to drive LPJ-GUESS.

In the maps, it is obvious that the analysis is applied to Australia, but for the spatially-averaged time-evolving diagrams, it is not always clear if they apply to Australia, globally, or some other area. Please make sure that that all captions are complete in stating what each diagram represents.

We thank the reviewer for their feedback and updated the figure captions accordingly.

Despite these quite severe caveats, I genuinely believe this can be turned in to a very useful manuscript and is appropriate to ESD. As I am not suggesting any additional analysis, then I am sure that a new version could be generated relatively quickly. There needs to be a rewrite that is much clearer as to what the analysis does (Australia only, multiple bias-correction methods for climate models, no temporal bias-correction and importantly, uncorrected LPJ-GUESS acting as "data"). And that, critically, will illustrate what the manuscript does not do (No corrections to LPJ-GUESS, no "data" as true vegetation carbon data e.g. from EO datasets, no

temporal bias-correction methods – and again, take note of the dangers of the later due to highly uncertain contemporary aerosol forcings).

We thank the reviewer for their detailed feedback and hope we addressed their concerns. We updated the title (see above) and figure captions to make clear that this analysis is focussed on Australia, clearly state that the bias correction methods applied do not correct temporal properties

> *and note that none of the correction methods used here are designed to correct temporal properties of the climate forcing.*

and explicitly mention the fact that simulated carbon variables (when LPJ-GUESS is forced with the CRUJRA reanalysis) are used as target datasets in this study in the introduction

> *We use a single dynamic global vegetation model, LPJ-GUESS (Smith et al., 2014), forced with different versions of CMIP6 climate forcing, as well as LPJ-GUESS forced with the CRUJRA reanalysis (Harris, 2019) as a target dataset for carbon variables, and focus on responses at seasonal to centennial timescales.*

and in the methods

> *In addition, we use LPJ-GUESS runs forced with the CRUJRA reanalysis as reference datasets for carbon variables.*

We further included a discussion point about the equilibrium climate sensitivity (see above).

---

## Author Response (AR3)

Dear editor,

We thank you for your assessment. To avoid confusion about the title of the manuscript (as pointed out by referee 3 and yourself), we revised the manuscript and included in the abstract that we are referring to a proverb

In this study,  *we open the proverbial Pandora's Box, and peer under the lid of strategies to tackle climate model ensemble uncertainty.*

We further explain the myth and link it to the narrative of our paper in the introduction

*Here, we examine multiple methods to constrain regional projections of the carbon cycle by opening Pandora's box, famous in Greek mythology. When Pandora could not resist opening the lid on her box, she allowed all the evils of the world to escape. Similarly, we could not resist testing the impact of various approaches to constraining the carbon cycle, and the challenges we identify are not easily resolvable. However, we hope that by highlighting these challenges we at least begin the process of resolving them.*

and in the implications

*To conclude, when Pandora opened the lid on her box she released the evils of the world, and these could never be put back into the box. We fear that we have also made the challenge of constraining the future regional-scale carbon budgets more difficult. We have, for example, raised more questions than answers, identified limitations of existing approaches and ultimately provided a challenge to the community to find more robust strategies to reduce the uncertainty in the projection of regional carbon stores. We acknowledge we have not provided easy answers, but we hope that by highlighting the challenges, strategies may be developed that can robustly constrain regional estimates of carbon storage.*

We hope that these adjustments address the concern about the title sufficiently.